

# Fidelity decay and error accumulation
# in random quantum circuits

Nadir Samos Sáenz de Buruaga[1★], Rafał Bistroń[2,3†], Marcin Rudziński[2,3],
Rodrigo M. C. Pereira[1], Karol Życzkowski[2,4] and Pedro Ribeiro[1]

**1** CeFEMA, LaPMET, Instituto Superior Técnico, Universidade de Lisboa,
Av. Rovisco Pais, 1049-001 Lisboa, Portugal
**2** Faculty of Physics, Astronomy and Applied Computer Science, Jagiellonian University,
ul. Łojasiewicza 11, 30-348 Kraków, Poland
**3** Doctoral School of Exact and Natural Sciences, Jagiellonian University,
ul. Łojasiewicza 11, 30-348 Kraków, Poland
**4** Center for Theoretical Physics, Polish Academy of Sciences,
Al. Lotników 32/46, 02-668 Warszawa, Poland

★ nadir.samos@tecnico.ulisboa.pt , † rafal.bistron@doctoral.uj.edu.pl

## Abstract

**Fidelity decay captures the inevitable state degradation in any practical implementation of a quantum process. We devise bounds for the decay of fidelity for a generic evolution given by a random quantum circuit model that encompasses errors arising from the implementation of two-qubit gates and qubit permutations. We show that fidelity decays exponentially with both circuit depth and the number of qubits raised to an architecture-dependent power and we determine the decay rates as a function of the amplitude of the aforementioned errors. Furthermore, we demonstrate the utility of our results in benchmarking quantum processors using the quantum volume figure of merit and provide insights into strategies for improving processor performance. These findings pave the way for understanding how states evolving under generic quantum dynamics degrade due to the accumulation of different kinds of perturbations.**

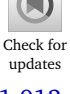
# 1   Introduction

Random quantum circuits (RQC) are an established tool to probe complex quantum dynamics, from disordered condensed matter setups to understanding quantum chaos and thermalization [1]. Their statistical properties, which can be analyzed using random matrix theory [2], crucially capture the behavior of complex yet structured phenomena.

A rather generic set of RQC is implemented by the circuit in Figs. 1 (a)-(b) which features $L$ qubits on a lattice where each layer $\tau = 1,\ldots,T$ implements a set of two-qubit random (Haar distributed) unitary operations [3] $u_{i,j}$, while $\Pi$ implements a given qubit permutation $P$ belonging to the symmetric group $S_L$. For future reference, we refer to this circuit configuration as *original*.

The role of permutations in RQC is to allow for direct interactions between any two degrees of freedom (qubits). When modeling complex dynamics, the choice of permutations reflects the geometry of the underlying structure. For example, the brickwork circuit—consisting of a

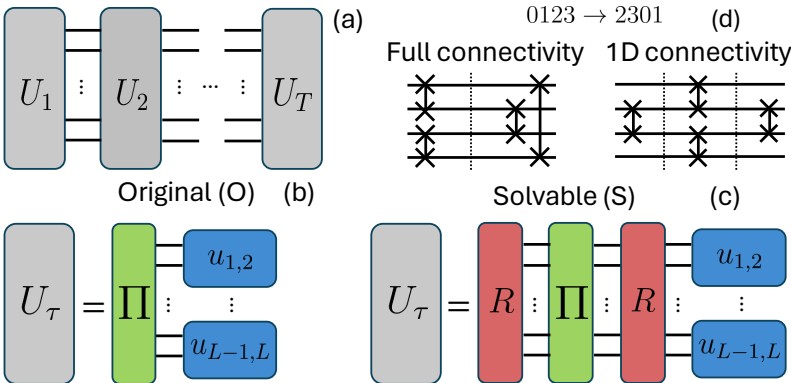

Figure 1: Diagram of the considered random circuit **(a)**. In the *original* circuit **(b)**, each layer of unitaries is composed of a random permutation and faulty 2-qubit gates. The *solvable* model **(c)** is obtained from the previous by acting with random global unitary gates $R$ before and after each permutation. Example of decomposition of the permutation $(0123) \to (2301)$ into SWAP gates **(d)** in fully connected and 1D architectures, requiring 2 and 3 sublayers, respectively.

sequential alternation of leftward and rightward single-qubit shifts—is a paradigmatic model of local random quantum circuits [4]. It has been studied extensively in the context of information spreading [5], thermalization [6], and measurement-induced phase transitions [1]. In contrast, random permutations have been used to model black hole dynamics with non-local interactions [7, 8], to establish bounds on entanglement generation [9, 10], to study pseudo-randomness and unitary $k$-designs—ensembles that reproduce Haar-random statistics up to the $k$-th moment [11, 12]—and to investigate quantum complexity [13], among other applications.

The versatility of RQCs makes then an ideal framework for addressing fundamental questions regarding universal dynamics. Practically, RQCs have proven valuable for probing quantum supremacy [14–16] and benchmarking noisy intermediate-scale [17] quantum processors [18–21]. Specifically, the well-established quantum volume (QV) test [22, 23], which roughly measures the maximum number of qubits that can be fully entangled, uses the layer configuration shown in Figs. 1 (a)-(b). Therefore, understanding the dynamic features of RQCs is highly desirable, given their fundamental importance and practical applications.

To model a generic quantum algorithm or general quantum dynamical processes, the permutations are uniformly sampled from $S_L$. However, in a quantum processor, each particular permutation is implemented by a decomposition into two-qubit gates that respect the processor's underlying architecture, i.e. the connectivity graph between qubits, as can be seen in Fig. 1 (d).

Here, we address the fundamental question of the stability of RQC dynamics against deviations from ideal unitary operations and how these affect the quantum state. At a practical level, answering these questions will also help us understand how errors accumulate in a non-ideal circuit implementation.

We rely on quantum fidelity, $\mathcal{F} = \langle\Psi|\tilde\rho|\Psi\rangle$, to quantify the deviations between the outcome state of the ideal quantum circuit $|\Psi\rangle$ and the outcome of physical realization $\tilde\rho$. In the following, $\tilde A$ denotes the perturbed version of the physical quantity $A$. Indeed, quantum fidelity has been a cornerstone concept in several areas, such as statistical mechanics [24, 25], quantum information [26, 27] and quantum chaos [28–30]. In quantum computing, quantum fidelity quantifies the error accumulation in the quantum processor and the consequent deviation between an ideal and a real outcome.

In the context of quantum chaos and dynamic phase transitions, quantum fidelity is termed Loschmidt echo [25, 31], measuring the extent to which a complex system is recovered after applying an imperfect (perturbed) time-reversal. In the framework of time-independent Hamiltonians, the behavior of the Loschmidt echo is well understood for single-particle quantum systems whose dynamics are fully chaotic in the classical limit: it typically exhibits an initial parabolic decay, followed by an exponential one, and eventually saturates [31]. This pattern has also been observed in many-body systems [32], and similar behaviour is expected in systems governed by time-dependent Hamiltonians, such as quantum circuits [33]. While a quantitative understanding is valuable in its own right, it becomes particularly pertinent in light of the technological relevance of quantum circuits.

The purpose of this work is to establish bounds on quantum fidelity for imperfect generic quantum evolution in RQC by analyzing its scaling with the circuit's width $L$ and depth $T$. We analyze the effects of implementing *faulty 2-qubit gates* affected by generic unstructured noise, and *faulty permutations* by assuming $\tilde{\Pi}$ is implemented by a combination of malfunctioning *swap* gates $\tilde{S}$ that interchange nearest-neighbor qubits within a specific architecture. We assume that the preparation of basis states and the final measurements are noiseless.

Our main result is an analytical expression for fidelity decay as a function of the number of qubits, layers and noise, that takes into account effective model architecture.

Furthermore, we demonstrate a practical application of this finding for benchmarking quantum processors by pinpointing the noise source mitigation of which most significantly improves the quantum volume metric. Additionally, we provide supporting evidence for the intuitive conjecture in Ref. [23] about the correlation between increased QV and connectivity. The appendices below the main text provide further details supporting our analytic and numerical results.

## 2 Setting the scene

In this section, we provide further details on the unitary errors considered in this work and the quantification of their impact through quantum fidelity.

### 2.1 Error modeling

For the original circuit in Fig. 1 (b), we model independently the errors that occur in permutations $\Pi_\tau$, and in the random two-qubit gates, $u_{r,r'}$, of the unitary layer $V_\tau = \bigotimes_{r=1}^{L/2} u_{2r-1,2r}$.

Due to architectural restrictions, each permutation has to be decomposed into available swap operations. Our error protocol assumes that the faulty swaps are implemented by a too short or too long pulse $S \to S^\beta$, where for each swap, the exponent $\beta$ is independently drawn from a Gaussian distribution with mean 1 and variance $\sigma^2$. After averaging over $\beta$, the faulty permutation operator, $\tilde{\Pi}$ can alternatively be reinterpreted as a composition of swaps $S$, each having a probability $p = \left[1 - \exp(-\pi^2\sigma^2/2)\right]/2$ of being omitted (see the proof in Appendix D). Since the same result can be obtained by averaging over $\beta$ or omitted swaps, we use the former in further discussion.

Since the two-qubit gates are already random, noise must be modelled as a random deviation from the uniform sampling defined by the Haar measure. To this end, to preserve the symmetry with which the gates were sampled we consider that each random unitary $u_{r,r'}$ is independently perturbed by unstructured noise: $\tilde{u}_{r,r'} = e^{i\alpha h_{r,r'}} u_{r,r'}$, where $h_{r,r'}$ is drawn from the Gaussian Unitary Ensemble (GUE), and $\alpha \geq 0$ controls the noise strength. Notably, the ability to model noise independently in both the permutations and the two-qubit gates provides significant flexibility and control in our framework.

## 2.2 Average quantum fidelity

The main object we analyze is the average fidelity $\overline{\mathcal{F}} = \overline{|\langle \Psi | \tilde{\Psi} \rangle|^2}$, between the ideal state

$$|\Psi\rangle = U_T U_{T-1}, \ldots, U_1 |\psi_0\rangle \,,$$

and the faulty one

$$\left|\tilde{\Psi}\right\rangle = \tilde{U}_T \tilde{U}_{T-1}, \ldots, \tilde{U}_1 |\psi_0\rangle \,,$$

where $\tilde{U}_\tau$ corresponds to the faulty, albeit unitary, realization of the faultless layer unitary $U_\tau$. For computing the average fidelity, it is useful to use the vectorized notation

$$\left|\psi^T\right\rangle = \langle \psi |^T \,,$$
$$\|\psi\phi\rangle = |\psi\rangle \otimes \left|\phi^T\right\rangle \,,$$

and work on a four-copy Hilbert space $\mathcal{H}(2^{4L})$. Hence we can write

$$|\langle \Psi | \tilde{\Psi} \rangle|^2 \to \left\langle +^{14;23} \right\| \tilde{\Psi}, \tilde{\Psi}^*, \Psi, \Psi^* \right\rangle \,,$$

where we define

$$\left\|+^{14;23}\right\rangle := \sum_{n m} \|mnnm\rangle \,, \qquad m = m_1, \ldots, m_L, \quad m_i = 0, 1 \,.$$

Equipped with this compact notation we obtain (see details in Appendix A)

$$\overline{\mathcal{F}} = \frac{1}{2^L} \left\langle +^{14;23} \right\| \prod_{\tau=1}^{T} \boldsymbol{\mathcal{U}} \left\| +^{1234} \right\rangle \,, \tag{1}$$

with

$$\boldsymbol{\mathcal{U}} = \overline{\left[ \tilde{U} \otimes \tilde{U}^* \otimes U \otimes U^* \right]}, \quad \text{and} \quad \left\|+^{1234}\right\rangle := \sum_n \|nnnn\rangle \,.$$

Hereafter, we use bold calligraphic typeface symbols, $\boldsymbol{\mathcal{U}}, \boldsymbol{\mathcal{V}}, \boldsymbol{\mathcal{R}}, \ldots$, to represent averaged super-operators acting on the four-copy Hilbert space. We have assumed that each layer is sampled independently and that the average over initial states is taken over all computational basis states. Expression (1) corresponds to the *entanglement fidelity* between perfect $\prod_\tau U_\tau$ and faulty $\prod_\tau \tilde{U}_\tau$ channels, estimation of which was analyzed in [34].

## 3 Solvable model

In the numerical simulations, we directly evaluated the average fidelity $\overline{\mathcal{F}(\alpha, p)}$ in terms of the two noise parameters. However such a calculation was not possible to perform analytically. Although each layer of 2-qubit gates is sampled independently, the presence of permutations shuffles the qubits, creating correlations between consecutive layers that prevent the average from factorizing as a product of independently averaged layers.

To overcome this challenge, we introduce a solvable model that exhibits a qualitative fidelity decay with $L$ and $T$ akin to the original circuit. The solvable circuit, presented in Fig. 1 (c) is derived from the original, presented in Fig. 1 (b) by introducing *faultless* $2^L \times 2^L$ Haar random unitaries $R_{2\tau-1}$ and $R_{2\tau}$ before and after each permutation $\Pi_\tau$, which in 4-copy formalism (1) leads to $\boldsymbol{\mathcal{R}} = \overline{R \otimes R^* \otimes R \otimes R^*}$. Due to the simple form of the average over $\boldsymbol{\mathcal{R}}$-layers [35] one can explicitly compute $\overline{\mathcal{F}}$ using Eq. (1).

It turns out that the difference between the average fidelity of the original and solvable models vanishes rapidly with $L$ and $T$. Intuitively, this can be attributed to the large mixing capacity of the original circuit. After a few layers of permutations braided with 2-qubit unitary gates, quantum states are so random that their properties are, on average, not influenced by the contribution of large random unitaries $R$. Thus, the introduction of $R$ does not affect the action of the next layers. Moreover, although the ability of two-qubit unitary layers to attain the degree of entanglement or expressibility of Haar distributed unitaries ($U(2^L)$) requires exponential deep circuits [4, 36, 37], the average fidelity can already be reproduced by a 2-design, thus we may expect relatively shallow circuits to approximate $U(2^L)$ features.

The outline of the calculation leading to the closed analytical form of the average fidelity decay of the solvable model is as follows. The average of each layer can be decomposed as $\mathcal{U} = \mathcal{R} \, \mathcal{P} \, \mathcal{R} \, \mathcal{V}$. The contribution of permutation $\mathcal{P}$, embedded by averages $\mathcal{R}$, can be expressed in terms of a global effective spin-1/2 orthonormal basis $\{\|\Uparrow\rangle, \|\Downarrow\rangle\}$, that remains invariant under $\mathcal{R}$ [1, 35]:

$$\mathcal{R} \, \mathcal{P} \, \mathcal{R} = \|\Uparrow\rangle \langle\Uparrow\| + \frac{\delta - 1}{4^L - 1} \|\Downarrow\rangle \langle\Downarrow\| , \tag{2}$$

where

$$\delta = \overline{(\mathrm{tr}\{\tilde{\Pi}(p)\Pi^T\})^2} = \overline{4^{m(P,p)}}, \tag{3}$$

is the $p$-dependent error factor. The last equality is obtained by a straightforward calculation (see Appendix D) where $m(P, p)$ is the number of cycles in the permutation $\tilde{P}(p)P^{-1}$, and $\tilde{P}(p)$ is the faulty implementation of permutation $P$ generated by our error protocol with error rate $p$. The contribution of faulty 2-qubit gates $\mathcal{V}$, is obtained by conveniently assembling each 2-qubit superoperator

$$\boldsymbol{u}_{2r-1,2r} = \|\Uparrow_{2r-1,2r}\rangle\langle\Uparrow_{2r-1,2r}\| + \frac{4f(\alpha) + 1}{5} \|\Downarrow_{2r-1,2r}\rangle\langle\Downarrow_{2r-1,2r}\| , \tag{4}$$

where the orthonormal basis $\{\|\Uparrow\rangle_{r,r'}, \|\Downarrow\rangle_{r,r'}\}$, spans the subspace of four-copy Hilbert space $\mathcal{H}(4^4)$ of two qubits. The function $f(\alpha)$ is closely related to the GUE(4) spectral form factor [38] and can be approximated by $f(\alpha) \approx e^{-5\alpha^2}$ in the relevant limit where the parametrized noise is small $\alpha \ll 1$ (see Appendix B).

Combining the contribution of both noise sources, we obtain the main result of this work, the formula for the average fidelity (see Appendix B)

$$\overline{\mathcal{F}} = \left(1 - \frac{1}{2^L}\right)\left(\frac{(\delta - 1)(\Delta - 1)}{(4^L - 1)^2}\right)^T + \frac{1}{2^L} . \tag{5}$$

The error factor $\delta$ is given in Eq. (3), while $\Delta = 2^L (3f(\alpha) + 1)^{L/2}$. The last term corresponds to the fidelity between two random $L$-qubit pure states [39].

## 4 Error accumulation

In this section, we present both numerical and analytical results on the impact of accumulating two-qubit and permutation errors across various architectures. We present compelling numerical evidence that the difference between the average fidelity of the original and solvable models vanishes rapidly with $L$ and $T$.

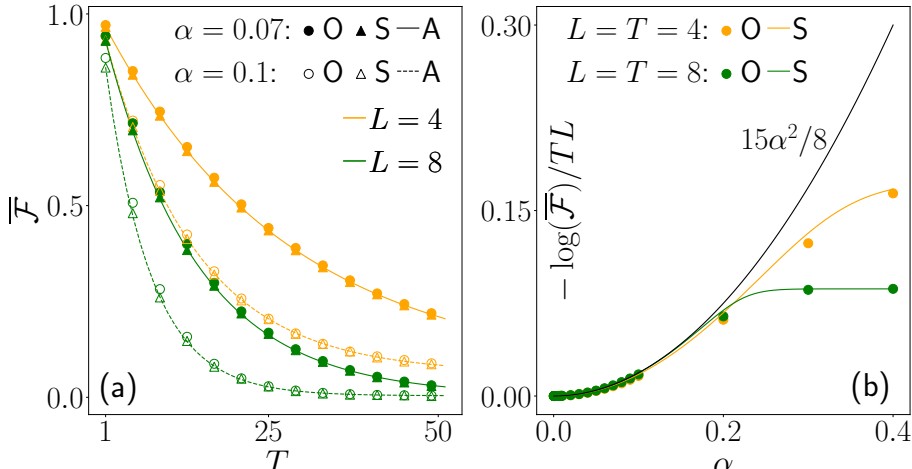

Figure 2: Fidelity evolution for the system sizes $L = 4$ and 8 for the *solvable* (S) and *original* (O) circuit together with the analytical prediction (A), for perfect permutations, $p = 0$. **(a)** Decay as a function of the number of layers $T$. Solid and dashed lines correspond to the analytical result for the (S) model with $\alpha = 0.07$ and $\alpha = 0.1$, respectively. **(b)** Exponent in fidelity decay as a function of the parameter $\alpha$ for $L = T = 4, 8$. Asymptotically, $\overline{\mathcal{F}} \approx e^{-15\alpha^2 LT/8}$. Observe that for sufficiently large $T$ or error strength $\alpha$, the average fidelity approaches $1/2^L$.

## 4.1 Faulty two-qubit gates

We first consider the scenario where only two-qubit gates have errors and permutations are noiseless, in which case $\delta(0) = 4^L$ in Eq. (5). For this case, Fig. 2 (a) and (b), show the fidelity as a function of the number of layers $T$ and noise level $\alpha$, respectively. In these and subsequent figures, the points represent numerical simulations of the original model, while the solid curves correspond to Eq. (5) obtained for the solvable model. Note the remarkable agreement between the two models even for small circuits. Fig. 2(b) also shows that, for small $\alpha$, the asymptotic for of Eq. (5) with $\delta = 4^L$,

$$\overline{\mathcal{F}} \sim e^{-\frac{15}{8}\alpha^2 LT}, \tag{6}$$

fits the data for both models.

Interestingly, the result in Eq. (6) is also found in brickwork circuit where the dynamics is local. A perturbative derivation of Eq. (6) for the brickwork circuit, valid for small $\alpha$, is given Appendix C, using a mapping to an effective 2D classical Ising model on a triangular lattice [5]. This result shows that the average fidelity exhibits universal decay across widely different models, further supporting our heuristic arguments justifying the validity of the solvable model.

## 4.2 Faulty permutations

In quantum processors, faulty permutations model architecture-specific connectivity errors. On the other hand, while simulating generic quantum dynamics, faulty permutations can be interpreted as lattice defects that modify how individual degrees of freedom interact.

### 4.2.1 Full connectivity

Consider the ideal processor in which each pair of qubits can be swapped in one move. In this case, each permutation $P$ can be decomposed into a set of cycles $\{C_i\}$ of $k_i$ elements,

and each cycle $C_i$ can be decomposed into $k_i - 1$ swaps combined in just two layers [40] (see Appendix D). Thus, permutations of $L$ qubits can be implemented using on average only $L - H_L \approx L - \log L - \gamma$ swaps, where $H_L \approx \log L + \gamma$ are Harmonic numbers describing the average number of cycles in permutation [40] and $\gamma$ is Euler's constant.

The property that for fully connected architectures any permutation can be implemented in two layers of different swaps implies that the omission of one swap in $\tilde{\Pi}(p)$ reduces exactly one cycle in $\tilde{P}(p)P^{-1}$. The error factor, $\delta(p)$, can be directly calculated (see Appendix D)

$$\delta_{\text{FC}}(p) \approx 4^L e^{-\frac{3}{4}p(L-\log L-\gamma)}, \tag{7}$$

by neglecting sub-leading corrections in higher powers of $p$. In the absence of two-qubit errors, i.e. $\alpha = 0$, this corresponds to a fidelity decay

$$\overline{\mathcal{F}}_{\text{FC}} \approx e^{-\frac{3}{4}pTL\left(1-\frac{\log L+\gamma}{L}\right)} \approx e^{-\frac{3}{4}pTL}, \tag{8}$$

in the limit of the large number of qubits $L$ and layers $T$.

### 4.2.2 1D architecture

Full connectivity is not feasible in practical scenarios and serves solely as a theoretical extreme case. For more general architectures, even optimal decompositions of permutations in terms of swaps' number, imply configurations where the omission of a given swap in $\tilde{\Pi}(p)$ may cancel previous errors, instead of introducing a new one. In the following, we focus on the regime of sparse errors when the frequency of error cancellation is negligible $p \ll 1$. In this case,

$$\delta \approx 4^L e^{-\frac{3}{4}p\overline{w(L)}}, \tag{9}$$

where $\overline{w(L)}$ is an average number of swaps needed to implement a permutation of $L$ qubits. With the increase of error probability $p$, this approximation deviates from the exact value of the error factor and starts serving as a lower bound (see Appendix E).

Before discussing higher-dimensional lattice structures, let us look at a one-dimensional qubit chain of size $L$. Here, a typical permutation $P$ changes the position of a qubit by an amount proportional to $L$. In this case, the decomposition of $P$ yielding the minimal number of swaps is obtained by the odd-even sort (also known as brick-sort or parity-sort algorithm) [41]. The number of swaps is given by a Mahonian distribution, with mean $w(L) = L(L-1)/4$, which quickly converges to Gaussian [42], yielding:

$$\delta_{d=1} \simeq 4^L e^{-\frac{3}{4}p\frac{L(L-1)}{4}}, \tag{10}$$

which, in the absence of other errors, corresponds to fidelity decay:

$$\overline{\mathcal{F}}_{d=1} \approx e^{-\frac{3}{16}pTL^2\left(1-\frac{1}{L}\right)} \approx e^{-\frac{3}{16}pTL^2}, \tag{11}$$

in the limit of the large number of qubits $L$ and layers $T$.

### 4.2.3 Other architectures

Although the exact optimal implementations are not known for cubic lattices with $d > 1$, the scaling in the number of swaps and layers required can be directly derived. As a simple example, in a 2D square lattice of $L$ qubits, the typical permutation displaces any qubit by a distance proportional to the length of a square $\sqrt{L}$. Therefore, allowing only nearest neighbour swaps, the average minimal number of swaps cannot be smaller than $\overline{w(L)} \propto L^{3/2}$. Furthermore, we have devised an algorithm to decompose permutations in cubic d-dimensional architectures

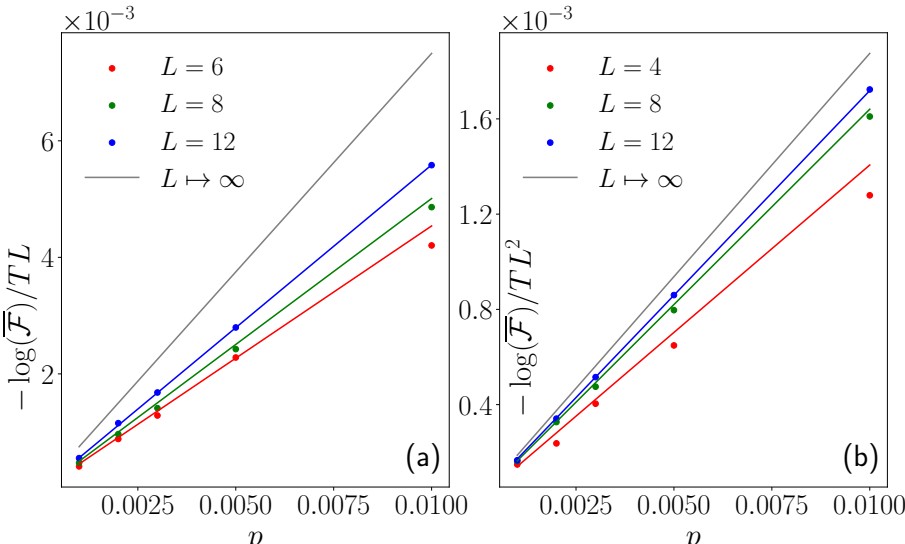

Figure 3: Fidelity decay as a function of swap omission probability $p$ for quantum volume circuits with perfect two-qubit gates $\alpha = 0$ with $L$ qubits and $T = L$ layers. **(a)** fully connected architecture, **(b)** linear architecture. To compare different numbers of qubits $L$ and layers $T$ the index in fidelity decay is divided by general trends. Lines correspond to theoretical bounds for a given number of qubits (8), (11), the asymptotic behaviour for many qubits is denoted by the grey line.

that has allowed us to derive upper bounds on $\overline{w(L)}$ as well (see Appendix E and implementation in [43]). From our algorithm, we obtain $\delta_d \gtrsim 4^L e^{-\frac{3}{4}(d-1/2)pL^{1+\frac{1}{d}}}$. In the absence of other errors, this leads to asymptotic bounds for the fidelity decay,

$$\overline{\mathcal{F}}_d > e^{-\frac{3}{4}(d-1/2)pTL^{1+1/d}}, \tag{12}$$

valid for a large number of qubits $L$ and layers $T$. In the case $d = 2$ one arrives at $\overline{\mathcal{F}}_{d=2} > e^{-\frac{9}{8}pTL^{3/2}}$.

## 5 Heavy output frequency vs fidelity

Our approach can be generalized to a more operational framework, directly related to *Quantum Volume* [22, 23]. This figure of merit relies on the heavy-output frequency $h_{\tilde{U}}$, which compares the outputs of faulty and ideal states represented in the computational basis. It is calculated as the average sum of probabilities measured in the experiment $\tilde{p}(m) = |\langle m|\tilde{\rho}|m\rangle|$, that exceed the median of the ideal distribution $p(m) = |\langle m|\Psi\rangle|^2$. The analyzed processor passes the Quantum Volume test for a given number of qubits when the average heavy output frequency is greater than $h_{\tilde{U}}^* = 2/3$ – see Ref. [14].

Fig. 4(a) (see also Appendix F) provides strong evidence that for the original circuit subjected to the noise models discussed, $\overline{\mathcal{F}}$ and $\overline{h_{\tilde{U}}}$ are linearly related. These results suggest that the linear dependence noted in the specific case of global depolarizing noise [44] is more general.

Plugging the limiting value of fidelity corresponding to $h_{\tilde{U}}^* = 2/3$ into the analytical expression Eq. (5) allows us to obtain the threshold line in the space of parameters $\alpha$ and $p$ below which a circuit of $L$ qubits passes the quantum volume test with QV equal to $2^L$. Fig. 4(b) shows this threshold line for two examples of two architectures, together with discrepancies obtained from the numerical study of the heavy output frequency on the original circuit.

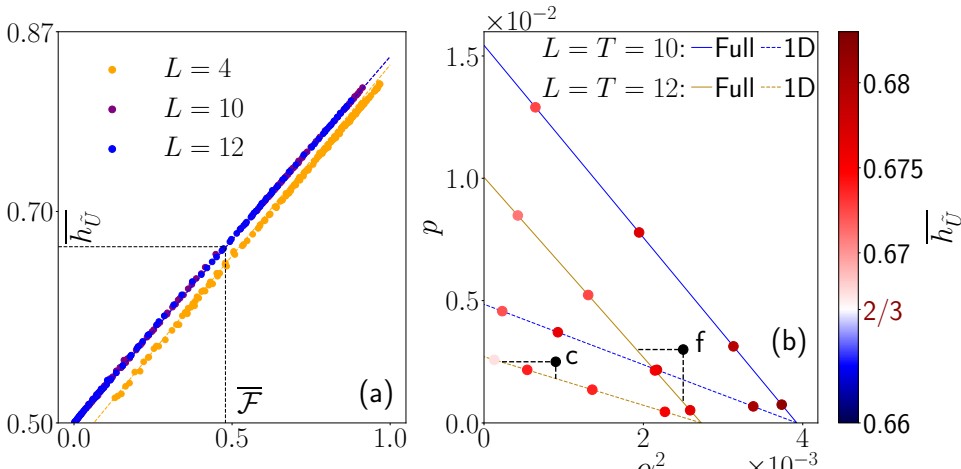

Figure 4: **(a)** Relations between average the heavy output frequency $\overline{h_{\tilde{U}}}$ and average fidelity $\overline{\mathcal{F}}$ for the *original* circuit (dots) and postulated linear relation (lines) for $12 \leq T \leq 20$, 1D and fully connected architectures, and different values of $\alpha$ and $p$. The threshold value $h_{\tilde{U}}^* = 2/3$ corresponds to a fidelity of $\mathcal{F}^* \approx 0.479$ (black dashed lines). **(b)** Threshold contour curves for $QV = 2^L$ for 1D (dashed line) and fully connected (full line) architectures. The coloured points represent the corresponding numerical values of $\overline{h_{\tilde{U}}}$ for the original circuit. Points f and c serve as representative quantum processors with full and 1D architectures, respectively. The QV of processor f improves equally by reducing either $\alpha^2$ or $p$, whereas for processor c reducing $p$ is more effective.

# 6 Conclusion

We quantify error accumulation for generic quantum dynamics implemented in a wide class of random quantum circuits. Unitary errors, arising from faulty permutations and defective two-qubit gates, are characterized by error parameters $p$ and $\alpha$, respectively. Our central result, in Eq. (5), offers an exact analytical prediction for fidelity decay when qubit connectivity follows a $d$-dimensional lattice structure. For a large number of qubits and layers, the average fidelity decays exponentially as

$$\overline{\mathcal{F}}_d \approx \exp\left(-\nu\alpha^2 LT\right)\exp\left(-\frac{3}{4}\mu(d)pL^{1+\frac{1}{d}}T\right), \tag{13}$$

where $\nu = 15/8$ is obtained using an effective solvable model, that approximates remarkably well all numerical results, and the lower bound $\mu(d) \leq d - \frac{1}{2}$ is established using our algorithm (see Appendix E and [43]).

This result further substantiates the claim of Ref. [23], that quantum computer architectures with higher connectivity yield substantially improved performances. Indeed, when errors affect two-qubit unitaries only ($p = 0$), the average fidelity decays in a universal manner across widely different circuit architectures.

By combining these predictions with our robust numerical findings of a linear relation between the average fidelity (13) and the heavy output frequency, our results can be operationally employed to access QV. Specifically, for a quantum processor characterized by noise parameters $(\alpha, p)$, we can directly compute its QV and identify the source of the limiting error. This insight can guide effective strategies for improving QV.

It is worth noting that in practice quantum processors often use classical optimization procedures to determine the most suitable way of implementing a given algorithm [44], such

as to minimize the number of gates or the use of error prone-qubits. For example, optimizing the decomposition of permutations into swaps within certain architectures – token swapping – has been extensively studied [40, 45–47]. However, these manipulations cannot decrease the order at which the number of two-qubit gates and swaps scale with $L$ and thus can only change the decay by a $p$ and $\alpha$-independent constant (see Appendix E). The architectures of existing quantum processors often correspond to rather irregular graphs. In those cases, our results continue to provide bounds for the fidelity decay given an estimated architecture dimensionality.

Presented results establish a general framework for understanding the degradation of quantum states in the implementation of a generic quantum evolution due to error accumulation. Specifically, this approach highlights how fidelity decay depends on errors from both local (two-qubit gates) and non-local (permutations) sources across varying levels of connectivity of the underlying architecture.

Although this work is limited to studying two of the most significant unitary errors, the versatility and agnostic nature of random circuits make the results presented here valuable for modeling modern quantum computers. Future work may focus on extending this framework to encompass other types of errors, including memory errors, crosstalk, and implementation-specific errors. In particular, non-unitary errors, which play a critical role in realistic quantum systems, are a key focus of our ongoing investigations. In addition, it could be interesting to explore the connection between average fidelity in the generic models considered and out-of-time order correlators (OTOCs), inspired by the known relation between the Loschmidt echo and OTOCs in systems governed by time-independent Hamiltonians [48].

## Acknowledgments

It is a pleasure to thank Lucas Sá, David Amaro Alcalá, Mateo Casariego, Ryszard Kukulski and Javier Molina-Villaplana for fruitful discussions and constructive remarks. We also acknowledge Tomaž Prosen and Sergiy Denisov for the early discussions that triggered the project.

**Funding information**   This work realized within the DQUANT QuantEra II Programme was supported by the National Science Centre, Poland, under the project 2021/03/Y/ST2/00193, and by FCT-Portugal Grant Agreement No. 101017733.[1] It received funding from the European Union's Horizon 2020 research and innovation programme under Grant Agreement No 101017733. PR, NB, and RP acknowledge further support from FCT-Portugal through Grant No. UID/CTM/04540/2020.

## A   Average fidelity computation

In this section, we derive the expression Eq. (1) of the main document. The faultless circuit yields a state $|\Psi\rangle$

$$|\Psi\rangle = U_T U_{T-1}, \ldots, U_1 |\psi_0\rangle\,, \tag{A.1}$$

whereas the more realistic circuit produces

$$\big|\tilde{\Psi}\big\rangle = \tilde{U}_T \tilde{U}_{T-1}, \ldots, \tilde{U}_1 |\psi_0\rangle\,, \tag{A.2}$$

---

[1] https://doi.org/10.54499/QuantERA/0003/2021.

where $\tilde{U}_\tau$ corresponds to the faulty realization of the clean layer unitary $U_\tau$. As discussed in the main text, we are interested in computing the average of the fidelity

$$\mathcal{F} = \left|\langle\Psi|\tilde{\Psi}\rangle\right|^2. \tag{A.3}$$

To do so, it is useful to use the vectorized notation.

$$\left|\psi^T\right\rangle = \langle\psi|^T, \tag{A.4}$$

$$\|\psi\phi\rangle = |\psi\rangle \otimes \left|\phi^T\right\rangle. \tag{A.5}$$

Equipped with it we can write the fidelity in a 4-copy Hilbert $\mathcal{H}(2^{4L})$ space:

$$\begin{aligned}
\mathcal{F} =& \left|\langle\Psi|\tilde{\Psi}\rangle\right|^2 \\
=& \langle\psi_0| U_1^\dagger \cdots U_T^\dagger \tilde{U}_T \cdots \tilde{U}_1 |\psi_0\rangle \langle\psi_0| \tilde{U}_1^\dagger \cdots \tilde{U}_T^\dagger U_T \cdots U_1 |\psi_0\rangle \\
=& \sum_{nm} \langle\psi_0| U_1^\dagger \cdots U_T^\dagger |m\rangle \langle m| \tilde{U}_T \cdots \tilde{U}_1 |\psi_0\rangle \langle\psi_0| \tilde{U}_1^\dagger \cdots \tilde{U}_T^\dagger |n\rangle \langle n| U_T \cdots U_1 |\psi_0\rangle \\
=& \sum_{nm} \langle mn\| \left(\tilde{U}_T \cdots \tilde{U}_1 \otimes \tilde{U}_T^* \cdots \tilde{U}_1^*\right) |\psi_0\rangle \otimes \left|\psi_0^T\right\rangle \langle nm\| U_T \cdots U_1 \otimes U_T^* \cdots U_1^* \|\psi_0\psi_0\psi_0\psi_0\rangle \\
=& \sum_{nm} \langle mnnm\| \overleftarrow{\prod_\tau} \left[\tilde{U}_\tau \otimes \tilde{U}_\tau^* \otimes U_\tau \otimes U_\tau^*\right] \|\psi_0\psi_0\psi_0\psi_0\rangle,
\end{aligned} \tag{A.6}$$

where $\leftarrow$ indicates the time order $T, T-1, \ldots, 1$ and

$$\sum_n |n\rangle = \left(\sum_{m=0,1} |m\rangle\right)^{\otimes L}.$$

The average fidelity is then

$$\overline{\mathcal{F}} = \sum_{nm} \langle mnnm\| \prod_{\tau=1}^T \overline{\left[\tilde{U} \otimes \tilde{U}^* \otimes U \otimes U^*\right]} \|\psi_0\psi_0\psi_0\psi_0\rangle = \sum_{nm} \langle mnnm\| \prod_{\tau=1}^T \mathcal{U} \|\psi_0\psi_0\psi_0\psi_0\rangle, \tag{A.7}$$

where we have assumed that each layer is sampled independently, so we can remove the subindex $\tau$.

It is convenient to introduce these two different notations

$$\sum_{nm} \|mnnm\rangle = \left\|+^{14;23}\right\rangle = \Big), \quad \text{and} \quad \sum_{nm} \|mmnn\rangle = \left\|+^{12;34}\right\rangle = \Big\rangle, \tag{A.8}$$

and to average the above quantity also with respect to all the all the computational basis states $|\psi_0\rangle = |m\rangle$

$$\sum_n \|nnnn\rangle = \left\|+^{1234}\right\rangle = \Big\}. \tag{A.9}$$

Hence, the final expression of the average fidelity is

$$\overline{\mathcal{F}} = \frac{1}{2^L} \left\langle+^{14;23}\right\| \prod_{\tau=1}^T \mathcal{U} \left\|+^{1234}\right\rangle. \tag{A.10}$$

# B  Average fidelity of solvable model circuit

In this section, we will obtain the expression Eq. (5) of the main document corresponding to the average fidelity of the solvable model introduced in the main text, where each layer of the solvable model consists of a faulty permutation $\tilde{\Pi}$ embedded between two faultless large unitaries $R_1, R_2 \in \text{CUE}(2^L)$ belonging to the *circular unitary ensemble*, that is, sampled uniformly according to the Haar measure, and a gate made of non-overlapping random faulty 2-qubit unitaries $\tilde{V} = \bigotimes_{r=1}^{L/2} \tilde{u}_{2r-1,2r}$. Therefore, we have the average of the superoperator $\mathcal{U}$ in Eq. (A.10) can be further decompose:

$$\mathcal{U} = \mathcal{V}\,\mathcal{R}\,\mathcal{P}\,\mathcal{R}, \tag{B.1}$$

where

$$\mathcal{V} = \overline{\tilde{V} \otimes \tilde{V}^* \otimes V \otimes V^*}, \tag{B.2}$$

$$\mathcal{P} = \overline{\tilde{\Pi} \otimes \tilde{\Pi} \otimes \Pi \otimes \Pi}, \tag{B.3}$$

$$\mathcal{R} = \overline{R \otimes R^* \otimes R \otimes R^*}. \tag{B.4}$$

On what follows, we will describe in detail each of the above quantities. First, the averages in the circular unitary ensemble have been widely studied and computed by means of Weingarten calculus [35, 49]. In particular, following the tensor network perspective introduced in [50], we have that given a random unitary $d \times d$ matrix $R \in \text{CUE}(d)$,

$$\mathcal{R} = \overline{R \otimes R^* \otimes R \otimes R^*} = \frac{1}{d^2-1}\left( \underset{\text{⊃⊂}}{\overset{\text{⊃⊂}}{}} + \Big)\Big( - \frac{1}{d}\left( \underset{\text{⊃}}{\overset{\text{⊃}}{}}\Big( + \Big)\underset{\text{⊂}}{\overset{\text{⊂}}{}} \right) \right), \tag{B.5}$$

where we have used the diagrammatic notation introduced above in Eq. (A.8).

Observe that the states $\left\Vert +^{14;23}\right\rangle$, $\left\Vert +^{12;34}\right\rangle$ and $\left\Vert +^{1234}\right\rangle$ are not orthogonal:

$$\left\langle +^{12;34} \middle\Vert +^{12;34}\right\rangle = \bigcirc \;=\; d^2 \;=\; \bigodot \;=\; \left\langle +^{14;23} \middle\Vert +^{14;23}\right\rangle,$$

$$\left\langle +^{12;34} \middle\Vert +^{14;23}\right\rangle = \; \Big) \;=\; d \;=\; \Big) \;=\; \left\langle +^{14;23} \middle\Vert +^{12;34}\right\rangle, \tag{B.6}$$

$$\left\langle +^{1234} \middle\Vert +^{14;23}\right\rangle = \; \Big) \;=\; d \;=\; \Big) \;=\; \left\langle +^{1234} \middle\Vert +^{12;34}\right\rangle. \tag{B.7}$$

Hence, we find an orthogonal basis via the Gram-Schmidt procedure:

$$\left\Vert \Uparrow\right\rangle = \frac{1}{d}\left\Vert +^{12;34}\right\rangle, \qquad \left\Vert \Downarrow\right\rangle = \frac{1}{\sqrt{d^2-1}}\left( \left\Vert +^{14;23}\right\rangle - \frac{1}{d}\left\Vert +^{12;34}\right\rangle \right). \tag{B.8}$$

On this basis, the expression Eq. (B.5) takes the simpler form

$$\mathcal{R} = \left\Vert \Uparrow\right\rangle\left\langle \Uparrow\right\Vert + \left\Vert \Downarrow\right\rangle\left\langle \Downarrow\right\Vert. \tag{B.9}$$

It is clear from the above expression that $\mathcal{R}$ is a projector since $\mathcal{R}^k = \mathcal{R}$, $k \in \mathbb{N}$. This property will be convenient in what follows.

## B.1  Computing $\mathcal{V}$

On this subsection we aim to compute the corresponding average contribution of the faulty 2-qubit gates

$$\mathcal{V} = \overline{\bigotimes_{r=1}^{L/2} \tilde{u}_{2r-1,2r} \otimes \bigotimes_{r=1}^{L/2} \tilde{u}_{2r-1,2r}^* \otimes \bigotimes_{r=1}^{L/2} u_{2r-1,2r} \otimes \bigotimes_{r=1}^{L/2} u_{2r-1,2r}^*}$$

$$= \bigotimes_{r=1}^{L/2} \overline{\tilde{u}_{2r-1,2r} \otimes \tilde{u}_{2r-1,2r}^* \otimes u_{2r-1,2r} \otimes u_{2r-1,2r}^*} = \bigotimes_{r=1}^{L/2} \boldsymbol{u}_{2r-1,2r}. \tag{B.10}$$

As discussed in the main text, each faulty gate is modeled as a random unitary $u_{r,r'} \in \text{CUE}(4)$ poissoned with an unstructured and generic noise given by a random unitary whose generator belongs to the *Gaussian unitary ensemble*, $\tilde{u}_{2r-1,2r} = e^{i\alpha h_{2r-1,2r}} u_{2r-1,2r}$ $h_{2r,2r-1} \in \text{GUE}(4)$, where $\alpha \geq 0$. Therefore, the local average of the 4-copied unitaries can be further decomposed.

$$\boldsymbol{u}_{2r,2r-1} = \overline{\tilde{u}_{2r-1,2r} \otimes \tilde{u}_{2r-1,2r}^* \otimes u_{2r-1,2r} \otimes u_{2r-1,2r}^*} \tag{B.11}$$

$$= \left( \overline{e^{i\alpha h_{2r-1,2r}} \otimes e^{-i\alpha h_{2r-1,2r}} \otimes \mathbb{1} \otimes \mathbb{1}} \right) \left( \overline{u_{2r-1,2r} \otimes u_{2r-1,2r}^* \otimes u_{2r-1,2r} \otimes u_{2r-1,2r}^*} \right).$$

We identify the rightmost average as the one outlined in the previous section $\mathcal{R}$ particularized for $d = 4$. Therefore, it remains to compute the average $\overline{e^{i\alpha h_{2r-1,2r}} \otimes e^{-i\alpha h_{2r-1,2r}}}$ with respect to the GUE measure. To keep the discussion more general, let us compute this quantity for an arbitrary $d$ random Hamiltonian $H \in \text{GUE}(d)$.

$$\overline{e^{i\alpha H} \otimes e^{-i\alpha H^*}} = \int dH e^{-\text{tr}\frac{H^2}{2}} \left( e^{-i\alpha H} \otimes e^{i\alpha H^*} \right), \qquad dH = \prod_{i=1}^{d} dH_{ii} \prod_{i>j} \frac{1}{\sqrt{2}} \mathfrak{Re} H_{ij} \mathfrak{Im} H_{ij}. \tag{B.12}$$

It is convenient to exploit the fact that the GUE measure is invariant under unitary transformations. In particular, this implies that the eigenvectors of $H$ do not favor any specific direction in Hilbert space. Therefore, the unitary matrix $U$ that diagonalizes $e^{i\alpha H}$, namely

$$e^{i\alpha H} = U D_\lambda U^\dagger, \quad \text{with} \quad D_\lambda = \text{diag}(e^{i\alpha\lambda_1}, \dots, e^{i\alpha\lambda_d}),$$

must be distributed according to the Haar measure. As a result, the average in Eq. (B.12) can be decomposed into an average over two copies of the Haar-distributed unitaries, as discussed above, together with an average over the eigenvalues encoded in $D_\lambda \otimes D_\lambda^*$, which must be performed with respect to the GUE joint probability distribution.

$$\overline{e^{i\alpha H} \otimes e^{-i\alpha H^*}} = \overline{(U \otimes U^*)} \overline{(D_\lambda \otimes D_\lambda^*)} (U^\dagger \otimes U^T) = \overline{(U \otimes U^*) \mathfrak{D} (U^\dagger \otimes U^T)}. \tag{B.13}$$

In terms of the diagrammatic notation, we can write:

$$\overline{e^{i\alpha H} \otimes e^{-i\alpha H^*}} = \overline{\left( \begin{array}{c} U \\ U^* \\ U^\dagger \\ U^T \end{array} \middle| \mathfrak{D} \right)} = \left( \begin{array}{c} U \\ U^* \\ U^* \\ U \end{array} \middle| \mathfrak{D} \right). \tag{B.14}$$

After interchanging rows three and four, we identify the first box as the average computed above $\mathcal{R}$ Eq. (B.5). Hence, it can be shown that

$$\overline{e^{i\alpha H} \otimes e^{-i\alpha H^*}} = \frac{1}{d^2-1} \left( \left( \boxed{\mathfrak{D}} - \frac{1}{d} \boxed{\mathfrak{D}} \right) \supset\subset + \left( \boxed{\mathfrak{D}} - \frac{1}{d} \boxed{\mathfrak{D}} \right) = \right). \tag{B.15}$$

The above expression can be further simplified taking into account

$$\left(\boxed{\mathcal{D}}\right) = \bigcirc = d\,, \quad \text{since} \quad D_\lambda D_\lambda^T = \mathbb{1}\,,$$

$$\left(\overline{\boxed{\mathcal{D}}}\right) = \left(\overline{\boxed{D_\lambda}}\right)^2 = \overline{\mathrm{tr}^2 D_\lambda} = \overline{\sum_{m,n=1}^{d} e^{i\alpha(\lambda_m - \lambda_n)}}\,. \tag{B.16}$$

We recognize in the second row the definition of the *spectral form factor* (SFF) [51, 52]: the Fourier transform of two point function, which is the simplest correlation function that displays universality. In this context, the strength of the noise $\alpha$ plays the role of time. For the GUE, there is a translational invariance between eigenvalues, and we can eliminate the specific dependence on $n, m$:

$$f_d(\alpha) := f_d(\lambda_n, \lambda_m, \alpha) = \int \prod_{i=1}^{d} d\lambda_i P_{\mathrm{GUE}}(\lambda_1, \dots, \lambda_d) e^{i\alpha(\lambda_m - \lambda_n)}$$

$$= \int \prod_{i=1}^{d} (d\lambda_i e^{-\frac{\lambda_i^2}{2}}) \prod_{\lambda_i > \lambda_j} (\lambda_i - \lambda_j)^2 e^{i\alpha(\lambda_m - \lambda_n)}\,, \qquad n \neq m\,, \tag{B.17}$$

where $P_{\mathrm{GUE}}$ is GUE probability density function. We postpone the computation of this expression for the next section. Now, it is enough to write the SFF as

$$\overline{\mathrm{tr}^2 D_\lambda} = d(1 + (d-1)f_d(\alpha))\,.$$

Thus, the average Eq. (B.12) is

$$\overline{e^{i\alpha H} \otimes e^{-i\alpha H^*}} = \frac{1 - f_d(\alpha)}{d+1} \,\supset\subset\, + \frac{d f_d(\alpha) + 1}{d+1} \,\overline{\phantom{-}}\,. \tag{B.18}$$

Next, we compute the product of the expression in Eq. (B.18) ( previously adding two identities/ horizontal lines to the diagrams) with the average over four unitaries as described in Eq. (B.5). Notice that specifying $d = 4$ results in the required $u_{2r-1,2r}$ as shown in Eq. (B.11), but for the sake of generality, we shall make this replacement at the end of the computation. Thus, we have:

$$\overline{(e^{i\alpha H} \otimes e^{-i\alpha H^*} \otimes \mathbb{1} \otimes \mathbb{1})(R \otimes R^* \otimes R \otimes R^*)}$$

$$= \frac{1}{d^2-1}\left\{ \frac{1-f_d(\alpha)}{d+1}\left[ \;\;\; + \;\;\; - \frac{1}{d}\left( \;\;\; + \;\;\; \right)\right]\right.$$

$$\left. + \frac{d f_d(\alpha)+1}{d+1}\left[ \;\;\; + \;\;\; - \frac{1}{d}\left( \;\;\; + \;\;\; \right)\right]\right\} \tag{B.19}$$

$$= \frac{1}{d^2-1}\left\{ \left(1 + \frac{f_d(\alpha)-1}{d(d+1)}\right) \;\;\; + \frac{d f_d(\alpha)+1}{d+1}\left( \;\;\; - \frac{1}{d}\;\;\; \left( \;\;\; - \frac{1}{d}\;\;\; \right) \right)\right\}\,,$$

where we have used the overlaps computed before Eq. (B.7). It is convenient to write the above expression in the state notation Eq. (A.8) and ommit the labeling of the gates, since the

above result is valid for all dimensions:

$$
\overline{(e^{i\alpha H} \otimes e^{-i\alpha H^*} \otimes \mathbb{1} \otimes \mathbb{1})(R \otimes R^* \otimes R \otimes R^*)}
\tag{B.20}
$$
$$
= \frac{1}{d^2-1}\left\{(1+\frac{f_d(\alpha)-1}{d(d+1)})\left\|+^{12;34}\right\rangle\left\langle+^{12;34}\right\|\right.
$$
$$
\left. + \frac{df_d(\alpha)+1}{d+1}\left(\left\|+^{14;23}\right\rangle\left\langle+^{14;23}\right\| - \frac{1}{d}\left\|+^{12;34}\right\rangle\left\langle+^{14;23}\right\| - \frac{1}{d}\left\|+^{14;23}\right\rangle\left\langle+^{12;34}\right\|\right)\right\}.
$$

In terms of the spin basis 1/2 introduced above, we invert Eq. (B.8):

$$
\left\|+^{12;34}\right\rangle = d\left\|\Uparrow\right\rangle, \qquad \left\|+^{14;23}\right\rangle = \sqrt{d^2-1}\left\|\Downarrow\right\rangle + \left\|\Uparrow\right\rangle.
\tag{B.21}
$$

Plugging the above in Eq. (B.20) yields

$$
\overline{(e^{i\alpha H} \otimes e^{-i\alpha H^*} \otimes \mathbb{1} \otimes \mathbb{1})(R \otimes R^* \otimes R \otimes R^*)}
\tag{B.22}
$$
$$
= \frac{1}{d^2-1}\left\{\left(1+\frac{f_d(\alpha)-1}{d(d+1)}\right)d^2\left\|\Uparrow\right\rangle\left\langle\Uparrow\right\|\right.
$$
$$
+ \frac{df_d(\alpha)+1}{d+1}\left((d^2-1)\left\|\Downarrow\right\rangle\left\langle\Downarrow\right\| + \left\|\Uparrow\right\rangle\left\langle\Uparrow\right\| + \sqrt{d^2-1}\left(\left\|\Downarrow\right\rangle\left\langle\Uparrow\right\| + \left\|\Uparrow\right\rangle\left\langle\Downarrow\right\|\right)\right)
$$
$$
\left. - \frac{d}{d}\sqrt{d^2-1}\left(\left\|\Uparrow\right\rangle\left\langle\Downarrow\right\| + \left\|\Downarrow\right\rangle\left\langle\Uparrow\right\|\right) - 2\frac{d}{d}\left\|\Uparrow\right\rangle\left\langle\Downarrow\right\|\right\}
$$
$$
= \frac{1}{d^2-1}\left(d^2 + \frac{df_d(\alpha)-d}{d+1} - \frac{df_d(\alpha)+1}{d+1}\right)\left\|\Uparrow\right\rangle\left\langle\Uparrow\right\| + \frac{1}{d^2-1}\left(\frac{df_d(\alpha)+1}{d+1}\left(d^2-1\right)\right)\left\|\Downarrow\right\rangle\left\langle\Downarrow\right\|
$$
$$
= \left\|\Uparrow\right\rangle\left\langle\Uparrow\right\| + \frac{df_d(\alpha)+1}{d+1}\left\|\Downarrow\right\rangle\left\langle\Downarrow\right\|.
\tag{B.23}
$$

Observe that the average is also diagonal in this basis, but the $\Downarrow$ sector depends on $f_4(\alpha)$, and consequently the projector property that the $\mathcal{R}$ possesses Eq. (B.9) is not valid anymore.

To conclude, we shall reintroduce the labelling of the unitaries and restrict the case for $d = 4$, yielding:

$$
\mathcal{V} = \bigotimes_{r=1}^{L/2}\left\|\Uparrow_{2r-1,2r}\right\rangle\left\langle\Uparrow_{2r-1,2r}\right\| + \frac{4f_4(\alpha)+1}{5}\left\|\Downarrow_{2r-1,2r}\right\rangle\left\langle\Downarrow_{2r-1,2r}\right\|.
\tag{B.24}
$$

In the remainder of this subsection, we compute explicitly $f_d(\alpha)$.

### B.1.1   Evaluating the function $f_d(\alpha)$

In this subsection we analyze the function $f_d(\alpha)$ introduced in Eq. (B.17), intimately connected with the spectral form factor. Whereas the large $L$ limit is well known; see, for instance, [2,38], the exact computation is less standard. Here we present a detailed derivation, similar to the one presented in [53], for arbitrary $d$ and $\alpha$, although in practice we are interested in $d = 4$ and $\alpha \ll 1$. This is, in a time-evolution Hamiltonian perspective, we are interested in the non-universal regime, where the nature of the matrix ensemble takes a great relevance. For a complete introduction to the random matrix, see, for example, [2,54]. Here we start by taking into account the definition of the n-point function:

$$
\rho^{(n)}(\lambda_1, \ldots, \lambda_n) = \int d\lambda_{n+1} \cdots d\lambda_d P_{\text{GUE}}(\lambda_1, \ldots, \lambda_d),
\tag{B.25}
$$

we see that

$$
f_d(\alpha) = \int \rho^{(2)}(\lambda_m, \lambda_n) e^{i\alpha(\lambda_m - \lambda_n)}, \qquad n \neq m.
\tag{B.26}
$$

The term $\prod_{\lambda_i > \lambda_j} (\lambda_i - \lambda_j)^2$ that appears with the Jacobian is called Vandermonde determinant $\Delta_d(\boldsymbol{\lambda})$:

$$\Delta_d^2(\boldsymbol{\lambda}) = (\boldsymbol{\lambda}) \prod_{\lambda_i > \lambda_j} (\lambda_i - \lambda_j)^2 = \det\left[ \{\lambda_j^{i-1}\}_{i,j=1}^d \right]^2 = \begin{vmatrix} 1 & 1 & \cdots & 1 \\ \lambda_1 & \lambda_2 & \cdots & \lambda_d \\ \vdots & \vdots & \vdots & \vdots \\ \lambda_1^{d-1} & \lambda_2^{d-1} & \cdots & \lambda_d^{d-1} \end{vmatrix}^2 . \tag{B.27}$$

Now we can harness the properties of the determinant. That is,

$$\Delta_d(\boldsymbol{\lambda}) = \det\left[ \{\lambda_j^{i-1}\}_{i,j=1}^d \right] = \det\left[ \{P_{i-1}(\lambda_j)\}_{i,j=1}^d \right], \tag{B.28}$$

where $P_k(x)$ $k = 0, \ldots, d-1$ are a family of monic orthogonal polynomials. In our case, the Hermite family is suitable. The reason is that they are orthogonal to the weight $e^{-\frac{x^2}{2}}$ that appears in our probability density function $P_{\text{GUE}}$:

$$\int_{-\infty}^{\infty} dx\, e^{-\frac{x^2}{2}} H_m(x) H_n(x) = \sqrt{2\pi} n! \delta_{mn} . \tag{B.29}$$

In order to play with orthonormal objects, we shall define the Hermite wavefunctions:

$$\Psi_m(x) = \frac{1}{(2\pi)^{1/4}} \frac{1}{\sqrt{m!}} e^{-\frac{x^2}{4}} H_m(x) . \tag{B.30}$$

Therefore, we write:

$$f_d(\alpha) = C \int \prod_{i=1}^d d\lambda_i \det\left[ \{e^{-\frac{\lambda_j^2}{4}} H_{i-1}(\lambda_j)\}_{i,j=1}^d \right] \det\left[ \{e^{-\frac{\lambda_j^2}{4}} H_{i-1}(\lambda_j)\}_{i,j=1}^d \right] e^{i\alpha(\lambda_m - \lambda_n)} \tag{B.31}$$

$$= (2\pi)^{d/2} \prod_{k=1}^{d-1} k! C \int \prod_{i=1}^d d\lambda_i \det\left[ \{\Psi_{i-1}(\lambda_j)\}_{i,j=1}^d \right] \det\left[ \{\Psi_{i-1}(\lambda_j)\}_{i,j=1}^d \right] e^{i\alpha(\lambda_m - \lambda_n)} , \qquad n \neq m .$$

Taking into account that $\det[A]\det[B] = \det[AB]$ and $\det[A] = \det[A^T]$:

$$f_d(\alpha) = (2\pi)^{d/2} \prod_{k=1}^{d-1} k! C \int \prod_{i=1}^d d\lambda_i \det\left[ \{\sum_{k=0}^{d-1} \Psi_k(\lambda_i) \Psi_k(\lambda_j)\}_{i,j=1}^d \right] e^{i\alpha(\lambda_m - \lambda_n)} \tag{B.32}$$

$$= C_d \int \prod_{i=1}^d d\lambda_i \det\left[ \{K_d(\lambda_i, \lambda_j)\}_{i,j=1}^d \right] e^{i\alpha(\lambda_m - \lambda_n)} , \qquad n \neq m , \tag{B.33}$$

where we have collected the constants, and

$$K_d(x,y) = \sum_{k=0}^{d-1} \Psi_k(x) \Psi_k(y) , \tag{B.34}$$

is the reproducing Hermite kernel. The reason is that it satisfies the property:

$$K_d(x,y) = \int_{-\infty}^{\infty} du\, K_d(x,u) K_d(u,y) . \tag{B.35}$$

This property allows the $d \times d$ determinant to be sequentially simplified by reducing its dimension. It can be shown that

$$C_d \int d\lambda_n \det\left[ \{K_d(\lambda_i, \lambda_j)\}_{i,j=1}^n \right] = C_d (K_d(\lambda_n, \lambda_n) - n + 1) \det\left[ \{K_d(\lambda_i, \lambda_j)\}_{i,j=1}^{n-1} \right], \tag{B.36}$$

with $K_d(\lambda_n, \lambda_n) = d$. Hence, iterating the above expression $d$ times, we obtain the normalization constant of the probability density function $C_d = 1/d!$, and we rewrite the 2-point function Eq. (B.25)

$$\rho^{(2)}(\lambda_m, \lambda_n) = \frac{(d-n)!}{d!} \begin{vmatrix} K_d(\lambda_m, \lambda_m) & K_d(\lambda_m, \lambda_n) \\ K_d(\lambda_n, \lambda_m) & K_d(\lambda_n, \lambda_n) \end{vmatrix}. \tag{B.37}$$

Using the above expression in Eq. (B.26) yields

$$\begin{aligned} f_d(\alpha) &= \frac{(d-2)!}{d!} \iint d\lambda_m d\lambda_n \begin{vmatrix} K_d(\lambda_m, \lambda_m) & K_d(\lambda_m, \lambda_n) \\ K_d(\lambda_n, \lambda_m) & K_d(\lambda_n, \lambda_n) \end{vmatrix} e^{i\alpha(\lambda_m - \lambda_n)} \\ &= \frac{1}{d(d-1)} \left( \left( \int dx K_d(x,x) e^{-i\alpha x} \right)^2 - \left( \int dx dy K_d(x,y) e^{-i\alpha(x-y)} \right)^2 \right). \end{aligned} \tag{B.38}$$

We have to compute the following integral:

$$\int_{-\infty}^{\infty} dx\, e^{-\frac{x^2}{2}} H_k(x) H_{k'}(x) e^{\pm i\alpha x}. \tag{B.39}$$

We make use of the following three results:

1. Given a function $f(x)$ that vanishes at infinity, the integral

$$\int_{-\infty}^{\infty} dx\, e^{-\frac{x^2}{2}} H_k f(x) = \int_{-\infty}^{\infty} dx\, e^{-\frac{x^2}{2}} f^{(k)}(x), \qquad f^{(k)}(x) = \frac{d^k}{dx^k} f(x), \tag{B.40}$$

   as can be seen by integrating $k$ times by parts.

2. The generalized Leibniz rule:

$$(f \times g)^{(k)} = \sum_{l=0}^{k} \frac{k!}{l!(k-l)!} f^{(k-l)} g^{(l)}. \tag{B.41}$$

3. The following recursion relation that the Hermite polynomials hold:

$$H_n^{(m)}(x) = \frac{n!}{(n-m)!} H_{n-m}(x). \tag{B.42}$$

To solve the integral Eq. (B.39), we first use the identity Eq. (B.40) with $f(x) = H_{k'}(x) e^{\pm i\alpha x}$, following by the Leibniz Eq. (B.41) rule, and finally Eq. (B.42):

$$\int_{-\infty}^{\infty} dx\, e^{-\frac{x^2}{2}} H_k(x) H_{k'}(x) e^{\pm i\alpha x} = \sum_{l=0}^{k} \frac{k! k'! (\pm i\alpha)^{k'-k+2l}}{l!(k-l)!(k'-k+l)!} \sqrt{2\pi} e^{-\frac{\alpha^2}{2}}. \tag{B.43}$$

We use the above result to obtain $f_d(\alpha)$. It is interesting to distinguish between the contribution of the connected and disconnected parts of the (Fourier transform of) the two-point function. The disconnected part is obtained by particularizing $k = k'$:

$$\begin{aligned} \left( \int_{-\infty}^{\infty} dx K_d(x,x) e^{\pm i\alpha x} \right)^2 &= \frac{1}{\sqrt{2\pi}} \sum_{k=0}^{d-1} \frac{1}{k!} \int_{-\infty}^{\infty} dx\, e^{-x^2/2} H_k^2(x) e^{-i\alpha x} \\ &= e^{-\alpha^2} \left( \sum_{k=0}^{d-1} k! \sum_{l=0}^{k} \frac{(\pm i\alpha)^{2l}}{(l!)^2 (k-l)!} \right)^2. \end{aligned} \tag{B.44}$$

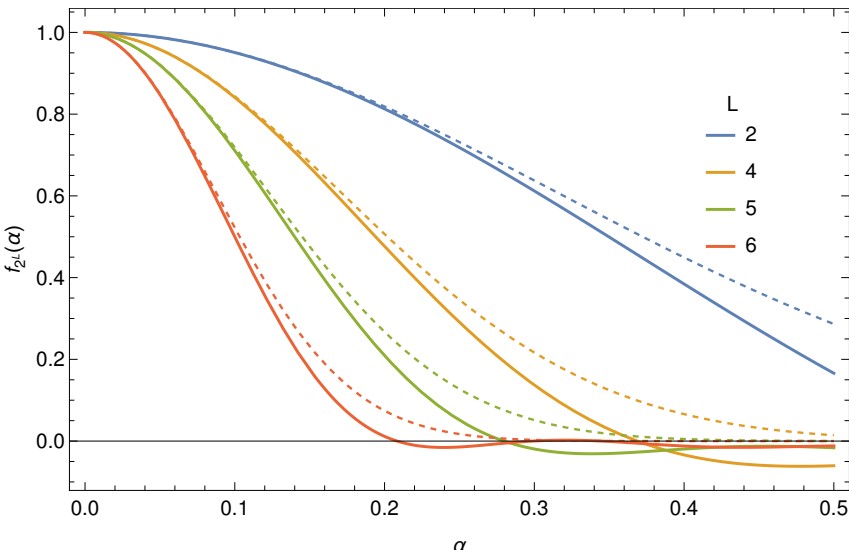

Figure 5: Plot of the function $f_d(\alpha)$ for $d = 4, 16, 32, 64$. The thick lines correspond to the analytic result Eq. (B.46), and the dashed lines to Eq. (B.48).

The connected part is:

$$\int_{-\infty}^{\infty} dx\, dy\, K_d(x,y) K_d(y,x) e^{\pm i\alpha(x-y)} \tag{B.45}$$

$$= \frac{1}{2\pi} \sum_{k=0}^{d-1} \sum_{k'=0}^{d-1} \frac{1}{k! k'!} \left( \int_{-\infty}^{\infty} dx\, e^{-x^2/2} e^{i\alpha x} H_k(x) H_{k'}(x) \right) \left( \int_{-\infty}^{\infty} dx\, e^{-y^2/2} e^{-i\alpha x} H_k(y) H_{k'}(y) \right)$$

$$= e^{-\alpha^2} \left( \sum_{k=0}^{d-1} \sum_{k'=0}^{d-1} k! k'! (-1)^{k'-k+1} \alpha^{k'-k} \sum_{l=l'=0}^{k} \frac{\alpha^{2(l+l')}}{l! l'! (k-l)! (k-l')! (k'-k+l)! (k'-k+l')!} \right).$$

So we arrive to the final result:

$$f_d(\alpha) = \frac{e^{-\alpha^2}}{d(d-1)} \left\{ \left( \sum_{k=0}^{d-1} k! \sum_{l=0}^{k} \frac{(\pm i\alpha)^{2l}}{(l!)^2 (k-l)!} \right)^2 \right.$$

$$\left. - \left( \sum_{k=k'=0}^{d-1} k! k'! (-1)^{k'-k+1} \alpha^{k'-k} \left| \sum_{l=0}^{k} \frac{(i\alpha)^{2l}}{l! (k-l)! (k'-k+l)!} \right|^2 \right) \right\}. \tag{B.46}$$

In particular, we find that

$$f_4(\alpha) = \frac{1}{36} e^{-\alpha^2} \left( -\alpha^{10} + \frac{25\alpha^8}{2} - 64\alpha^6 + 138\alpha^4 - 144\alpha^2 + 36 \right). \tag{B.47}$$

Remarkably, it can be seen that $f_d(\alpha) = 1 - (d+1)\alpha^2 + O(\alpha^4)$, so we shall propose a more manageable expression:

$$f_d(\alpha) \sim e^{-(d+1)\alpha^2}. \tag{B.48}$$

The validity of this approximation can be examined in Fig. 5

## B.2 Computing $\mathcal{R} \, \mathcal{P} \, \mathcal{R}$

Now we shall focus on the average of the faulty permutation contribution $\mathcal{P}$. The large $d \times d$ unitaries that *encapsulate* the permutation contribution allow us to assume that the effect of

each permutation is decorrelated, making for a great simplification. As a first step, we harness the left/right invariance of the Haar measure:

$$\mathcal{R}\,\mathcal{P}\,\mathcal{R} = \left(\overline{R\otimes R^* \otimes R \otimes R^*}\right)\left(\tilde{\Pi}\otimes\tilde{\Pi}\otimes\Pi\otimes\Pi\right)\left(\overline{R\otimes R^* \otimes R \otimes R^*}\right)$$

$$= \left(\overline{R\otimes R^* \otimes R \otimes R^*}\right)\left(\tilde{\Pi}\Pi^T\otimes\tilde{\Pi}\Pi^T\otimes\mathbb{1}\otimes\mathbb{1}\right)\left(\overline{R\otimes R^* \otimes R \otimes R^*}\right).$$

Hence,

$$
\mathcal{R}\,\mathcal{P}\,\mathcal{R} = \frac{1}{(d^2-1)^2}\left[\left(\supset\atop\supset - \frac{1}{d}\,\supset\!\supset\right)\right)\,\subset\atop\subset + \left(\supset\atop\supset - \frac{1}{d}\,\supset\!\supset\right)\left(\subset\atop\subset\right)\right]
$$
$$
\times\ \boxed{\Xi\otimes\Xi}\ \times\ \left[\supset\atop\supset\left(\subset\atop\subset - \frac{1}{d}\,\subset\!\subset\right) + \supset\atop\supset\left(\subset\!\subset - \frac{1}{d}\,\subset\atop\subset\right)\right],
$$

(B.49)

where $\Xi = \tilde{\Pi}\Pi^T$. Using Eq. (B.16) we find that

$$
\boxed{\Xi\otimes\Xi} = d\left(\boxed{\Xi\otimes\Xi}\right) = d^2\,,
$$

$$
\boxed{\Xi\otimes\Xi} = \boxed{\Xi\otimes\Xi} = \overline{\text{tr}^2\,\tilde{\Pi}\Pi^T} := \delta\,,
$$

(B.50)

$$
\boxed{\Xi\otimes\Xi} = \boxed{\Xi\otimes\Xi} = \left(\boxed{\Xi\otimes\Xi}\right) = d\,.
$$

Therefore, we find that

$$
\mathcal{R}\,\mathcal{P}\,\mathcal{R} = \frac{1}{(d^2-1)^2}\left\{\left(d^2 - 2 + \frac{\delta}{d^2}\right)\supset\!\subset\atop\supset\!\subset + (\delta - 1)\left(\supset\atop\supset\right)\left(\subset - \frac{1}{d}\,\supset\atop\supset\right)\left(\subset - \frac{1}{d}\,\subset\atop\supset\right)\subset\!\subset\right\}.
$$

(B.51)

Finally, we write it in terms of the orthogonal basis Eq. (B.8):

$$
\mathcal{R}\,\mathcal{P}\,\mathcal{R} = \|\Uparrow\rangle\,\langle\Uparrow\| + \frac{\delta - 1}{d^2 - 1}\,\|\Downarrow\rangle\,\langle\Downarrow\|\,.
$$

(B.52)

Recall that $\delta = \overline{\text{tr}^2\,\tilde{\Pi}\Pi^T}$. In the next section, we give details regarding the calculation of this average and the nature itself of the faulty permutation $\tilde{\Pi}$.

## B.3   Putting all together

We have all the ingredients to compute the average fidelity Eq. (A.10), but they need to be written on a common basis: while $\mathcal{R}\mathcal{P}\mathcal{R}$ is written in terms of the total spins $\Uparrow, \Downarrow$, $\mathcal{V}$ is written in terms of local spins $\Uparrow_{2r-1,2r}, \Downarrow_{2r-1,2r}$ with $r = 1,\ldots,L/2$ as can be seen in Eq. (B.24). To do so, we introduce the notation $\|m,i\rangle$ where $m$ counts the number of spins $\Downarrow_{2r-1,2r}$ and $i$ labels the degeneracy. We shall write all the quantities, namely $\|\Uparrow\rangle, \|\Downarrow\rangle, \mathcal{U}$ in terms of this basis.

The easiest starting point is $\|\Uparrow\rangle$, since

$$\|\Uparrow\rangle = \otimes_{r=1}^{L/2} \left\|\Uparrow_{2r-1,2r}\right\rangle = \|0,1\rangle \,. \tag{B.53}$$

The state $\|\Downarrow\rangle$ is a complicated combination consequence of being a superposition state (see Eq. (B.8)) Equivalently (see Eq. (B.8))

$$\begin{aligned}
\|\Downarrow\rangle &= \frac{1}{\sqrt{4^L-1}}\left(\bigotimes_{r=1}^{L/2}\left\|+_{2r-1,2r}^{\mathbf{14;23}}\right\rangle - \frac{1}{2^L}\bigotimes_{r=1}^{L/2}\left\|+_{2r-1,2r}^{\mathbf{12;34}}\right\rangle\right) \\
&= \frac{1}{\sqrt{4^L-1}}\left(\bigotimes_{r=1}^{L/2}\left(\left\|\Uparrow_{2r-1,2r}\right\rangle + 15^{1/2}\left\|\Downarrow_{2r-1,2r}\right\rangle\right) - \|0,1\rangle\right) \\
&= \frac{1}{\sqrt{4^L-1}}\sum_{m=1}^{L/2} 15^{m/2}\sum_{i=1}^{\binom{L/2}{m}}\|m,i\rangle \,,
\end{aligned}$$

where in the second equality we have inverted Eq. (B.8) and particularized for $d = 4$. It is important to note that the summation index starts from 1 and not from 0 in the final result.

Now we express $\mathcal{V}$ Eq. (B.24) and $\mathcal{R}\,\mathcal{P}\,\mathcal{R}$ Eq. (B.52) in the new basis $\{\|m,i\rangle\}$:

$$\begin{aligned}
\mathcal{V} &= \bigotimes_{r=1}^{L/2}\left\|\Uparrow_{2r-1,2r}\right\rangle\left\langle\Uparrow_{2r-1,2r}\right\| + \frac{4f_4(\alpha)+1}{5}\left\|\Downarrow_{2r-1,2r}\right\rangle\left\langle\Downarrow_{2r-1,2r}\right\| \\
&= \sum_{m=0}^{L/2}\left(\frac{4f_4(\alpha)+1}{5}\right)^m\sum_{i=1}^{\binom{L/2}{m}}\|m,i\rangle\,\langle m,i\| \,, \\
\mathcal{R}\,\mathcal{P}\,\mathcal{R} &= \|\Uparrow\rangle\,\langle\Uparrow\| + \frac{\delta-1}{d^2-1}\|\Downarrow\rangle\,\langle\Downarrow\| \\
&= \|0,1\rangle\,\langle 0,1\| + \left(\frac{\delta-1}{4^L-1}\right)\left(\frac{1}{4^L-1}\right)\sum_{m,n=1}^{L/2} 15^{\frac{m+n}{2}}\sum_{i,j=1}^{\binom{L/2}{m}\binom{L/2}{n}}\|m,i\rangle\,\langle n,j\| \,.
\end{aligned} \tag{B.54}$$

Observe that the summation index in $\mathcal{V}$ runs from 0 in this case. Hence, we are finally able to write the average of one layer:

$$\mathcal{U} = \mathcal{V}\,\mathcal{R}\,\mathcal{P}\,\mathcal{R} \tag{B.55}$$

$$= \left(\sum_{m=0}^{L/2}\left(\frac{4f_4(\alpha)+1}{5}\right)^m\sum_{i=1}^{\binom{L/2}{m}}\|m,i\rangle\,\langle m,i\|\right)$$

$$\times\left(\|0,1\rangle\,\langle 0,1\| + \left(\frac{\delta-1}{4^L-1}\right)\left(\frac{1}{4^L-1}\right)\sum_{n,p=1}^{L/2} 15^{\frac{n+p}{2}}\sum_{j,k=1}^{\binom{L/2}{n}\binom{L/2}{p}}\|n,j\rangle\,\langle p,k\|\right) \tag{B.56}$$

$$= \|0,1\rangle\,\langle 0,1\| + \left(\frac{\delta-1}{4^L-1}\right)\left(\frac{1}{4^L-1}\right)\sum_{m,n=1}^{L/2} 15^{\frac{m+n}{2}}\left(\frac{4f_4(\alpha)+1}{5}\right)^m\sum_{i,j=1}^{\binom{L/2}{m}\binom{L/2}{n}}\|m,i\rangle\,\langle n,j\| \,,$$

where we have used that $\langle m,i\,\|n,j\rangle = \delta_{m,n}\delta_{i,j}$. Indeed, we are able to exponentitate the

above quantity thanks to the fact that the crossed terms vanish:

$$(\mathcal{U})^T = \|0,1\rangle\langle 0,1\| + \left[\left(\frac{\delta-1}{4^L-1}\right)\left(\frac{1}{4^L-1}\right)\sum_{m=1}^{L/2}15^m\left(\frac{4f_4(\alpha)+1}{5}\right)^m\binom{L/2}{m}\right]^{T-1}$$

$$\times\left[\left(\frac{\delta-1}{4^L-1}\right)\left(\frac{1}{4^L-1}\right)\sum_{m,n=1}^{L/2}15^{\frac{m+n}{2}}\left(\frac{4f_4(\alpha)+1}{5}\right)^m\sum_{i,j=1}^{\binom{L/2}{m}\binom{L/2}{n}}\|m,i\rangle\langle n,j\|\right]. \quad (B.57)$$

To finally find the average fidelity, we only need to compute the overlaps $\langle+^{14;23}\|\mathcal{U}^T\|+^{1234}\rangle$. First,

$$\langle+^{14;23}\|0,1\rangle = \frac{1}{2^L}\langle+^{14;23}\|+^{12;34}\rangle = 1,$$

where we have used Eq. (B.53) and Eq. (B.7). It is straightforward to check that

$$\langle+^{14;23}\|\left(\sum_{m=1}^{L/2}15^{\frac{m}{2}}\left(\frac{4f_4(\alpha)+1}{5}\right)^m\sum_{i=1}^{\binom{L/2}{m}}\|m,i\rangle\right) \quad (B.58)$$

$$= \sum_{m,n=1}^{L/2}15^{\frac{m+n}{2}}\left(\frac{4f_4(\alpha)+1}{5}\right)^m\sum_{i,j=1}^{\binom{L/2}{m}\binom{L/2}{n}}\langle n,j\|m,i\rangle = \sum_{m=1}^{L/2}15^m\left(\frac{4f_4(\alpha)+1}{5}\right)^m\binom{L/2}{m}.$$

Also, from Eq. (B.7) and Eq. (B.8) it is clear that

$$\langle 0,1\|+^{1234}\rangle = \frac{1}{2^L}\langle+^{12;34}\|+^{1234}\rangle = 1, \quad (B.59)$$

and that $\langle\Downarrow_{2r-1,2r}\|+^{1234}_{2r-1,2r}\rangle = 3/\sqrt{15}$. Then we can fin the last overlap:

$$\left(\sum_{m=1}^{L/2}15^{\frac{m}{2}}\sum_{j=1}^{\binom{L/2}{m}}\langle m,j\|\right)\|+^{1234}\rangle = \sum_{m=1}^{L/2}3^m\binom{L/2}{m}. \quad (B.60)$$

Hence, we finally obtain:

$$\overline{\mathcal{F}} = \frac{1}{2^L}\left(1 + \left[\left(\frac{\delta-1}{4^L-1}\right)\left(\frac{1}{4^L-1}\right)\left(\sum_{m=1}^{L/2}15^m\left(\frac{4f_4(\alpha)+1}{5}\right)^m\binom{L/2}{m}\right)\right]^T\left(\sum_{m=1}^{L/2}3^m\binom{L/2}{m}\right)\right)$$

$$= \frac{1}{2^L}\left(1 + \left[\left(\frac{\delta-1}{4^L-1}\right)\frac{\left(15\frac{4f_4(\alpha)+1}{5}+1\right)^{L/2}-1}{4^L-1}\right]^T(2^L-1)\right)$$

$$= \left(1-\frac{1}{2^L}\right)\left[\left(\frac{\delta-1}{4^L-1}\right)\left(\frac{(12f_4(\alpha)+4)^{L/2}-1}{4^L-1}\right)\right]^T + \frac{1}{2^L}. \quad (B.61)$$

Defining $\Delta(\alpha) = 2^L(3f_4(\alpha)+1)^{L/2}$, we get to the final result Eq. (4) of the main text.

# C  Faulty gates in brickwork architecture

On this section, we shall consider brickwork architecture, which is a popular random circuit considered as a toy model of simulating black holes, chaotic evolution, and monitored setting, among others [1]. The main trait that differentiate it from the *original* circuit, or quantum volume circuit that we considered in the main text is that it implements a well defined notion of spatial locality. Of course, it can be seen as a particular realization of the original circuit, with two kind of fixed permutations:

$$\Pi_{\text{even}} = \begin{pmatrix} 1 & 2 & 3 & \cdots & L \\ L & 1 & 2 & \cdots & L-1 \end{pmatrix}, \qquad \Pi_{\text{odd}} = \begin{pmatrix} 1 & 2 & 3 & \cdots & L \\ 2 & 3 & 4 & \cdots & 1 \end{pmatrix}. \tag{C.1}$$

Since we want to preserve the notion of locality, we only consider faulty gates. Then, using Eqs. (A.10) and (B.10) we have:

$$\overline{\mathcal{F}} = \left\langle +^{\mathbf{14;23}} \middle\| (\mathcal{V}_{\text{even}} \mathcal{V}_{\text{odd}})^{T/2} \middle\| +^{\mathbf{1234}} \right\rangle = \left\langle +^{\mathbf{14;23}} \middle\| \left[ \left( \bigotimes_{r=1}^{L/2} \mathbf{u}_{2r-1,2r} \right) \left( \bigotimes_{r=1}^{L/2} \mathbf{u}_{2r,2r+1} \right) \right]^{T/2} \middle\| +^{\mathbf{1234}} \right\rangle, \tag{C.2}$$

where we implicitly assumed cyclic boundary conditions $L+1 \equiv 1$. As before, we could write the even and odd contributions terms of the spin $1/2$ basis Eq. (B.24) $\left\| \sigma_{2r-1,2r} \right\rangle$ and $\left\| \sigma_{2r,2r+1} \right\rangle$ respectively where $\sigma = \uparrow, \downarrow$. However, the states are not orthogonal, and consequently, the computation becomes quite involved. From the previous section, Eq. (B.20), we have:

$$\begin{aligned}
\mathbf{u}_{rr'} &= \frac{1}{d^2 - 1} \bigg\{ \left( 1 + \frac{f_d(\alpha) - 1}{d(d+1)} \right) \left\| +^{\mathbf{12;34}}_{rr'} \right\rangle \left\langle +^{\mathbf{12;34}}_{rr'} \right\| \\
&\quad + \frac{d f_d(\alpha) + 1}{d+1} \left( \left\| +^{\mathbf{14;23}}_{rr'} \right\rangle \left\langle +^{\mathbf{14;23}}_{rr'} \right\| - \frac{1}{d} \left\| +^{\mathbf{12;34}}_{rr'} \right\rangle \left\langle +^{\mathbf{14;23}}_{rr'} \right\| - \frac{1}{d} \left\| +^{\mathbf{14;23}}_{rr'} \right\rangle \left\langle +^{\mathbf{12;34}}_{rr'} \right\| \right) \bigg\} \\
&= \left\| \uparrow_{rr'} \right\rangle \left\langle +^{\mathbf{12;34}}_{rr'} \right\| + \left\| \downarrow_{rr'} \right\rangle \left\langle +^{\mathbf{14;23}}_{rr'} \right\|,
\end{aligned} \tag{C.3}$$

particularized for $d = 4$, and where we have introduced two new (only ket) states:

$$\begin{aligned}
\left\| \uparrow_{rr'} \right\rangle &= \frac{1}{300} \left( (19 + f_4(\alpha)) \left\| +^{\mathbf{12;34}}_{rr'} \right\rangle - (4 f_4(\alpha) + 1) \left\| +^{\mathbf{14;23}}_{rr'} \right\rangle \right), \\
\left\| \downarrow_{rr'} \right\rangle &= \frac{4 f_4(\alpha) + 1}{75} \left( \left\| +^{\mathbf{14;23}}_{rr'} \right\rangle - \frac{1}{4} \left\| +^{\mathbf{12;34}}_{rr'} \right\rangle \right).
\end{aligned} \tag{C.4}$$

To simplify further the calculation, we can represent Eq. (C.3) as a sum of two diagrams:



Equipped with them, we can assemble them yielding a brickwork setup as the original circuit as shown in Fig. 6 (left). However, observe that, despite this similarity, it is conceptually different, since, each box represents a four-copy deterministic (averaged) superoperator, instead of a random gate. Furthermore, we can calculate the contractions between the states $\left\langle +^s_{rr'} \middle\| \sigma \right\rangle$, with $s = \mathbf{12;34}$ or $s = \mathbf{14;23}$, of the layer $\tau - 1$, and $\sigma = \uparrow, \downarrow$ of layer $\tau$. Notice that the above contractions correspond to integrating out the spins $+^s_{rr'}$, retaining only the spins $\sigma_r$ placed on the vertices of a triangular lattice, as the red triangles in Fig. 6 (left) indicate. Hence, we have a 2D classical Ising model [5] of $L/2$ spins on a triangular lattice of $T - 1$ columns. Each triangular plaquette is described by the (unnormalized) Boltzmann weights between the spin neighbors $\Delta^{\sigma_3}_{\sigma_1, \sigma_2}$. All of them are listed in Fig. 6 (right). For example, the condition $\Delta^{\downarrow}_{\uparrow, \uparrow} = 0$ indicates the impossibility of configurations where two $\uparrow$ spins are at the base of the triangle with a $\downarrow$ spin at the apex.

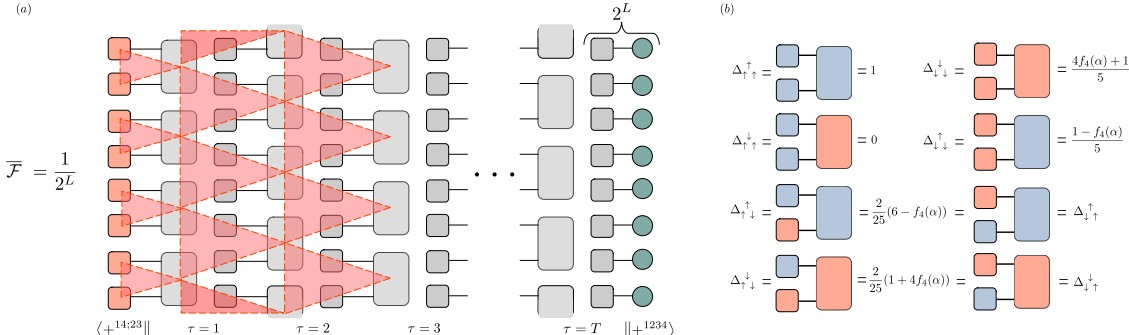

Figure 6: (a): Triangular lattice of a spin $1/2$ statistical model $L/2$ ($L = 8$ in this case) spins. The left boundary is completely fixed, whereas the right boundary is free, yielding a $2^L$ factor during overlap with the final column.(b): Boltzmann weights obtained from Eq. (C.4).

Finally, notice that the computation of the fidelity Eq. (C.2) corresponds to the fixation of the boundary conditions on the lattice and accouting for all potential configurations $\mathcal{Z}$ that adhere to these conditions. According to Eq. (C.5), the left boundary *injects* spins $\downarrow$, while the right boundary is free since following Eq. (B.7), $\left\langle +_{r,r'}^{\mathbf{1234}} \middle\| +_{rr'}^{\mathbf{14;23}} \right\rangle = \left\langle +_{r,r'}^{\mathbf{1234}} \middle\| +_{rr'}^{\mathbf{14;23}} \right\rangle = 4$. Let us first consider the situation with no error $\alpha = 0$. Taking into account Fig. 6 (b) and that $f(0) = 1$ (see Eq. (B.46)), the Boltzmann weights are more symmetrical (see [5]): $\Delta_{\uparrow\uparrow}^{\uparrow} = \Delta_{\downarrow\downarrow}^{\downarrow} = 1$, whereas $\Delta_{\uparrow\uparrow}^{\downarrow} = \Delta_{\downarrow\downarrow}^{\uparrow} = 0$, and finally $\Delta_{\uparrow\downarrow}^{\uparrow} = \Delta_{\downarrow\uparrow}^{\uparrow} = \Delta_{\uparrow\downarrow}^{\downarrow} = \Delta_{\downarrow\uparrow}^{\downarrow} = 2/5$. Since the left boundary forces all spins to be $\downarrow$, and there is no possibility of generating a *defect* $\uparrow$ ($\Delta_{\downarrow\downarrow}^{\uparrow} = 0$), all spins in the lattice are fixed to be $\downarrow$. Therefore, the only weight involved in all plaquettes is $\Delta_{\downarrow\downarrow}^{\downarrow} = 1$, and the right boundary produces a factor $2^L$ that conveniently cancels the denominator, so we arrive at $\overline{\mathcal{F}} = 1$.

The situation becomes more interesting and involved when $\alpha \neq 0$. Since we are interested in the small error regime $\alpha \ll 1$, taking into account Eq. (B.46) and expanding the $f_4(\alpha)$ to leading order, we can write the weights as

$$\Delta_{\uparrow\uparrow}^{\uparrow} = 1 \quad ; \quad \Delta_{\uparrow\uparrow}^{\downarrow} = 0,$$
$$\Delta_{\downarrow\downarrow}^{\downarrow} \approx 1 - 4\alpha^2 \quad ; \quad \Delta_{\downarrow\downarrow}^{\uparrow} \approx \alpha^2, \tag{C.6}$$
$$\Delta_{\uparrow\downarrow}^{\uparrow} = \Delta_{\downarrow\uparrow}^{\uparrow} \approx \frac{2}{5}(1 - 4\alpha^2) \quad ; \quad \Delta_{\uparrow\downarrow}^{\downarrow} = \Delta_{\downarrow\uparrow}^{\downarrow} \approx \frac{2}{5}(1 + 5\alpha^2).$$

Since we are interested on leading order, it is clear that only one plaquette can weigh $\Delta_{\downarrow\downarrow}^{\uparrow} = \alpha^2$: Providing a dynamical interpretation where the spins rearrange from left to right, a pair of $\downarrow$ spins can result in a single $\uparrow$ spin at the apex. Hence, we can write the average fidelity as

$$\overline{\mathcal{F}} \approx \mathcal{Z}_0 + \mathcal{Z}_1, \tag{C.7}$$

where

$$\mathcal{Z}_0 = (\Delta_{\downarrow\downarrow}^{\downarrow})^{LT/2} = 1 - 2LT\alpha^2 + O(\alpha^4), \tag{C.8}$$

and $\mathcal{Z}_1$ is the sum of all configurations with only one creation of a defect, i.e. just one $\Delta_{\downarrow\downarrow}^{\uparrow} = \alpha^2$ on one of the possible triangles.

Hence, the next step in the calculation is to count all such possible configurations. This is not a trivial task since even though we only allow the creation of one defect, once it is created, it can spread and reproduce towards the right boundary since $\Delta_{\uparrow\downarrow}^{\uparrow} = \Delta_{\downarrow\uparrow}^{\uparrow} = \Delta_{\uparrow\downarrow}^{\downarrow} = \Delta_{\downarrow\uparrow}^{\downarrow} \approx \frac{2}{5}$.

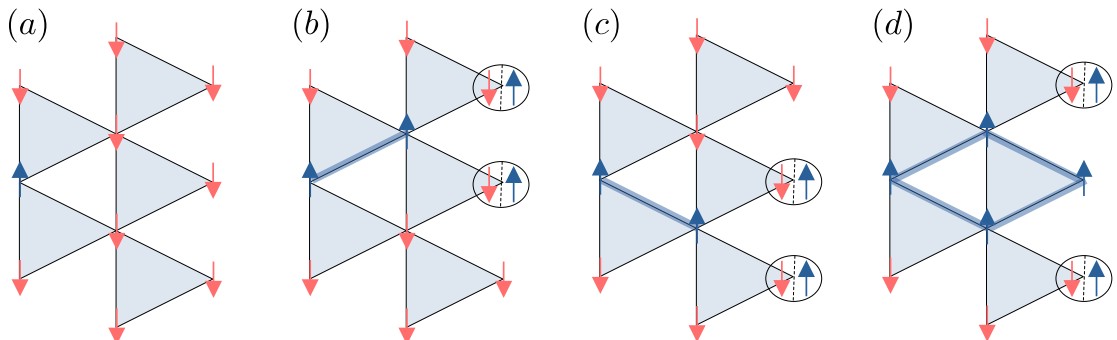

Figure 7: Scheme of the possible configurations at times $\tau$ and $\tau + 1$ originated by the appearence of a defect at $\tau - 1$. Discussion in the main text.

To see this, suppose that a defect is created in column $\tau - 1$ with cost $\alpha^2$. Then, all the vertices of the basis of the triangles of column $\tau$ are $\downarrow$ except one. Then, we can distinguish four possible scenarios schematized in Fig. 7:

(a) The defect *dies* with a cost of $(2/5)^2$ if the affected triangles are $\Delta_{\downarrow\uparrow}^{\downarrow}$ and $\Delta_{\uparrow\downarrow}^{\downarrow}$.

(b) (c) The defect spreads with a cost $(2/5)^2$ either up $\Delta_{\downarrow\uparrow}^{\uparrow}\Delta_{\uparrow\downarrow}^{\downarrow}$ or down $\Delta_{\downarrow\uparrow}^{\downarrow}\Delta_{\uparrow\downarrow}^{\uparrow}$ to the right.

(d) The defect reproduces and moves both up and down to the right with a cost

$$\Delta_{\downarrow\uparrow}^{\uparrow}\Delta_{\uparrow\uparrow}^{\uparrow}\Delta_{\uparrow\downarrow}^{\uparrow} = (2/5)^2.$$

Observe that all spins outside the light cone of the defect are $\downarrow$ and $\Delta_{\downarrow\downarrow}^{\downarrow} \approx 1$, and that all the above posibilities weight the same. Thus, it is convenient to introduce the notion of *length* of the defect, which corresponds to the number of layers that the propagation of the defect lasts until it dies, and it represented with the blue line in the figures. Hence, we can further decompose

$$\mathcal{Z}_1 = \sum_{\ell=0}^{T} \mathcal{Z}_1(\ell). \tag{C.9}$$

Then $\mathcal{Z}_1(\ell)$ contains all the states with a defect that appears at the apex of one of the triangles of column $\tau$, spreads following Fig. 7 (b),(c) and (d) until $\tau + \ell$, meaning that a situation Fig. 7(a) is reached at $\tau + \ell + 1$. Consider as a warm-up the case $\ell = 0$. The defect can be created at $\tau = 1, \ldots T - 2$ and die because the triangles at $\tau + 1$ are those of case (a). In addition, the defect can be injected into the boundary (see Fig. 6 (a)). Finally, the defect can be created in the last layer $T - 1$, without dying since it lies in the right (free) boundary. Putting all together:

$$\mathcal{Z}_1(0) = \alpha^2 \frac{L}{2}\left((T-1)\left(\frac{2}{5}\right)^2 + 1\right) + O(\alpha^4). \tag{C.10}$$

Consider now the case $\ell = 1$. We make the following considerations. First, only cases (b) and (c) yield a length defect $\ell = 1$. The case (d) always produces a defect of at least $\ell + 1$. Second, the last possible column to harbor the defect is at $T - 2$, and third, there are three possible configurations ((b), (c), and (d)) at the boundary. Hence, we have:

$$\mathcal{Z}_1(1) = \alpha^2 \frac{L}{2}\left((T-2)\left(\frac{2}{5}\right)^4 2 + \left(\frac{2}{5}\right)^2 3\right) + O(\alpha^4). \tag{C.11}$$

Now we shall write the partition function of defect configurations of length $\ell$ originating

Table 1: All possible diagrams that account for the possible configrations of the domain wall in Fig. 7. The Catalan numbers $\chi_{\ell+1}$ can be obtained by counting the number of diagrams of each column $\ell$, whereas the number of total configurations $\xi_\ell$ is obtained by counting all diagrams with the same time steps to be produced. This is indicated with the little number next to each diagram.

| $\ell = 0$ | $\ell = 1$ | $\ell = 2$ | $\ell = 3$ |
|---|---|---|---|

only from a single insertion ($\Delta_{\downarrow\downarrow}^{\uparrow}$). Suppose first that case (d) is not possible, and the only possibilities are that the defect propagates to the right, either moving up or down. This can be seen as a *zigzagging snake* of length $\ell$ if we study the movement of the domain wall, represented by the blue bold lines in Fig. 7. Thus, each snake weigths $(2/5)^{2\ell+1}$ and we have to count all of them. Since at each time it can move up or down, there are $2^\ell$ different snakes. The situation is more complicated when we include the case (d) which describes the bifurcation of the snake. Fortunately, this only affects the total number of such configurations $\chi_\ell$, since the only spins that contribute nontrivially are those located in the boundaries of the propagation (see Fig. 7 (d)). In terms of the domain, at each step it can: move up, move down one unit length, or bifurcate, and consequently jump two length units. In Table 1 we represent the first configurations until length $\ell = 2$. It can be seen that $\chi_\ell = 1, 2, 5, 14, \dots$ which corresponds to the Catalan numbers [55]

$$\chi_m = \frac{(2m)!}{(m+1)!m!},\tag{C.12}$$

for $m = \ell + 1$.

Furthermore, we must count all the configurations $\xi_\ell$ of defects of length $\ell$ that touch the right boundary without room to die, and consequently weight differently $(2/5)^{2\ell}$. We find $\xi_\ell = 1, 3, 10, 35, 126, \dots$ [56], which can be obtained by counting the diagrams with the same number in Table 1, and it can be written as:

$$\xi_\ell = \binom{2\ell+1}{\ell+1}.\tag{C.13}$$

Hence, putting all together we obtain:

$$\mathcal{Z}_1(\ell) = \alpha^2 \frac{L}{2}\left((T-1-\ell)\left(\frac{2}{5}\right)^{2(\ell+1)}\chi_{\ell+1} + \left(\frac{2}{5}\right)^{2\ell}\xi_\ell\right) + O(\alpha^4),\tag{C.14}$$

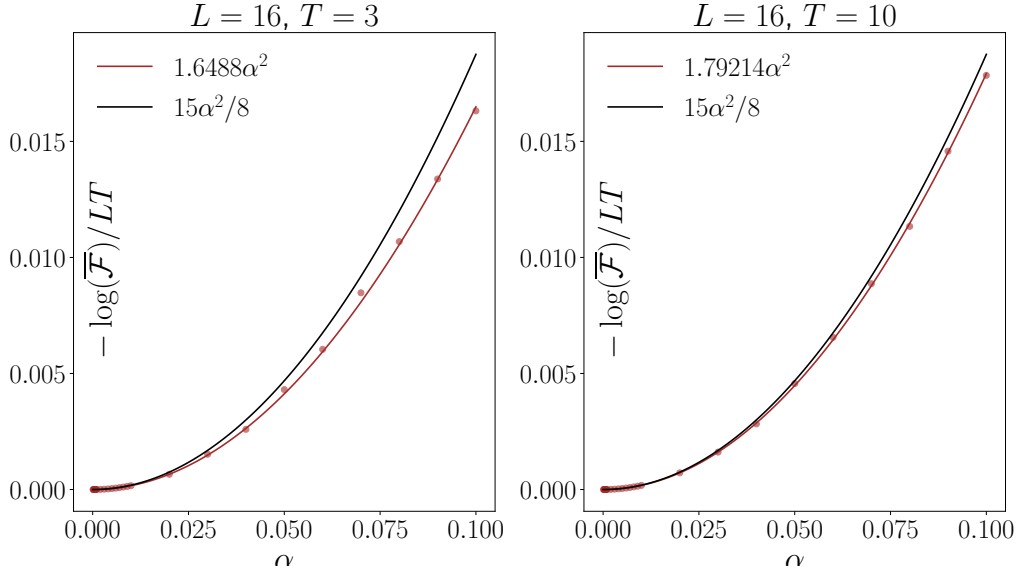

Figure 8: Average fidelity of the brickwork circuit with faulty gates. The brown curves correspond to the prediction of Eq. (C.15). Observe that the approximation of the solvable model becomes better as we consider deeper circuits.

Hence, the average fidelity for small error Eq. (C.7) is

$$
\overline{\mathcal{F}} = 1 - \alpha^2 L \left( 2T - \frac{1}{2} \sum_{\ell=0}^{T-1} (T-1-\ell) \left(\frac{2}{5}\right)^{2(\ell+1)} \chi_{\ell+1} + \left(\frac{2}{5}\right)^{2\ell} \xi_\ell \right)
$$
$$
= 1 - \alpha^2 \left[ \frac{15}{8} LT - L \left( \frac{5}{6} - \left(\frac{4}{5}\right)^{2T} \frac{9}{25\sqrt{\pi}} \Gamma\left(T+\frac{3}{2}\right) {}_2\tilde{F}_1\left(2, T+\frac{3}{2}; T+2; \frac{16}{25}\right) \right) \right], \quad \text{(C.15)}
$$

where ${}_2\tilde{F}_1(a; b; c; d)$ is the regularized Gauss hyper-geometric function.

Hence, observe that for $LT \gg 1$,

$$
\overline{\mathcal{F}} \approx 1 - \frac{15}{8} LT \alpha^2, \quad \text{(C.16)}
$$

which is the same value that we obtain by taking the Taylor expansion of the average fidelity of the solvable model Eq. (B.61).

For shallow circuits, the finite-size terms are also relevant. In Fig. 8 we show the average fidelity for small error $\alpha$. We also plot the prediction given by the asymptotic case Eq. (C.16), Observe that it improves as we consider deeper circuits, which is convenient, since Quantum Volume circuits are square.

To wrap up this section, it is important to recognize that the feasibility of this computation stems from the uniform arrangement of the permutation matrices that form the brickwork, which are faultless. The presence of faulty permutations significantly complicates the adaptation to the statistical model, making the approach based on a solvable model much more advantageous.

# D   Permutations implementation

In this section, we discuss the structure of permutations and their implementation. Our results then let us establish the analytical formula for the error factor $\delta$ of permutation for the fully connected architecture as a starting point to for discussing outer architectures.

## D.1 Imperfect SWAP gates

We address the imperfections in SWAP gate operations, characterizing each swap $S$ in a permutation $\Pi$ as imprecise due to variations in impulse duration. Such an model of error is natural, if one uses $\sqrt{S}$ as the universal 2-qubit gate (instead of, for example, CNOT), in which case the $S$ gate is a simple composition of two fundamental gates $S = \sqrt{S}\sqrt{S}$ and each of them could be performed imperfectly.

We consider an error model in which faulty implementation of $S$ of the form $S \to S^\beta$, where $\beta$ follows a Gaussian distribution with a mean of 1 and variance $\sigma$, independently for each swap. Since swap operation can be decomposed as $S = P_S - P_A$, where $P_S$ is a projection onto symmetric subspace and $P_A$ is a projection onto an antisymmetric subspace, the faulty implementation possess similar form $S^\beta = P_S + e^{i\pi\beta}P_A$. For any density matrix $\rho$ the average over imperfect swapping reads

$$\int_{-\infty}^{\infty} d\beta \, S^\beta \rho (S^\beta)^\dagger \frac{1}{\sigma\sqrt{2\pi}} e^{-\frac{(\beta-1)^2}{2\sigma^2}}$$

$$= \int_{-\infty}^{\infty} d\beta \, \left[(P_S\rho P_S) + (P_A\rho P_S)e^{i\pi\beta} + (P_S\rho P_A)e^{-i\pi\beta} + (P_A\rho P_A)\right] \frac{1}{\sigma\sqrt{2\pi}} e^{-\frac{(\beta-1)^2}{2\sigma^2}}$$

$$= (P_S\rho P_S) - (P_A\rho P_S)e^{-\frac{1}{2}\pi^2\sigma^2} - (P_S\rho P_A)e^{-\frac{1}{2}\pi^2\sigma^2} + (P_A\rho P_A) = p\rho + (1-p)\,S\rho S\,.$$

Which turns out to simplify into a probabilistic mixture with swapping probability $1-p$ and no swapping probability $p$, where

$$p = \frac{1}{2}(1 - e^{-\frac{1}{2}\pi^2\sigma^2})\,. \tag{D.1}$$

Thus from now on, we will use such an integrated scenario keeping in mind the original source of errors. Since the model of imperfect swapping directly corresponds to the probabilistic occurrence of swaps with $p \in [0, 1/2]$ and larger values of $p$ are futile from the perspective of quantum computer performance, we restrict ourselves to $p$ to such a range.

This model allows us to perform various estimations of the error factor $\delta(p) = \delta$ as in (B.50). In the simplest one of those, when we consider the implementation of a generic permutation of $L$ qubit system by expanding in the following series (averaging over all permutations of $L$ elements)

$$\delta(p) = \frac{1}{L!}\sum_{P\in\Sigma_L}\langle\text{Tr}[\tilde{\Pi}(p)\Pi^\top]^2\rangle_p = \frac{1}{L!}\sum_{P\in\Sigma_L}(1-p)^{w(P)}\text{Tr}[\Pi\,\Pi^\top]^2 + \ldots > 4^L(1-p)^{w(P)}, \tag{D.2}$$

where $P$ is a permutation of $L$ qubits, $\Pi$ is the corresponding operator and $\tilde{\Pi}(p)$ is a faulty realization of $\Pi$. In the second step, we extracted the cases with all swaps executed properly. Note that the only parameters are the probability $p$ of not performing a swap and the number of swaps $w$ necessary to implement a permutation.

The analytical expressions for the number of swaps $w$ necessary to implement each permutation are not known for most architectures. In such cases, we may further bound Eq. (D.2) further in the following way. Let $m_w$ be a number of permutations which demand $w$ swaps to implement, then:

$$\delta(p) \geq \frac{4^L}{L!}\sum_{P\in\Sigma_L}(1-p)^w = \frac{4^L}{L!}\sum_w m_w(1-p)^w \geq \frac{4^L}{L!}\sum_w m_w(aw+b)\,. \tag{D.3}$$

Where $f(w) = aw + b$ is a tangent line to a function $(1-p)^w$ in a point with $w$ equal its average over all permutations $w = \overline{w}$. More precisely, $a = \ln(1-p)(1-p)^{\overline{w}_L}$,

$b = (1 - \ln(1-p)\overline{w}_L)(1-p)^{\overline{w}_L}$. Because $f(w) = aw + b$ is tangent to a convex function, the last inequality in (D.3) is obvious. The remaining calculations are quite simple:

$$\delta(p) \geq \frac{4^L}{L!} \sum_w m_w(aw + b) = \frac{4^L}{L!} \left( a \sum_w m_w w + b \sum_w m_w \right) = \frac{4^L}{L!}(a\, L!\, \overline{w}_m + b\, L!)$$
$$= a\overline{w} + b = 4^L(1-p)^{\overline{w}_L} = 4^L e^{-q\overline{w}_L}, \tag{D.4}$$

with $q = -\ln(1-p) \approx p \approx \frac{\pi^2 \sigma^2}{4}$ for small $p$. Therefore to know a lower-bound one just has to know the average number of swaps.

## D.2  Fully connected architecture

In this section we provide a detailed computation of Eq. (6) of the main text. We consider a fully connected quantum architecture that allows arbitrary qubit interactions. In this model, any permutation can be decomposed into at most $L$ swap operations. The decomposition process involves organizing the permutation into cycles, where swaps within different cycles can be executed simultaneously. Each cycle can be transformed into 2-cycles using one layer of swaps, with the transformation process described as follows: starting with any two adjacent elements, subsequent swaps are performed between the next nearest elements until the cycle is completed. Since a 2-cycle can be reduced to the identity with a single swap, every permutation can be implemented with exactly two layers. See Fig. 9, decomposition of cycle $C$, and for more more detailed discussion see [40].

Moreover, because each cycle of $m$ elements involves only $m-1$ swaps, the number of necessary swaps equals $L$ minus the number of cycles. Fortunately, the average number of cycles is given by Harmonic numbers $H_L = \sum_{k=1}^{L} \frac{1}{k} \approx \ln L + \gamma$, where $\gamma \approx 0.577$ is the Euler–Mascheroni constant. Therefore the average number of necessary swaps grows as

$$\overline{w}_L = L - H_L \approx L - (\ln L + \gamma), \tag{D.5}$$

with the dimension $L$.

More precisely, the distribution of the permutations of $L$ elements with $m$ cycles is given by the Stirling numbers $\begin{bmatrix} L \\ m \end{bmatrix}$. The number of swaps is given by $k = L - m$ since for each cycle one swap can be omitted, as argued before.

To derive the formula for the error factor $\delta(p)$ (Eq. (3) of the main text), we aim to translate the number of omitted swaps in $\tilde{P}(p)$ into the number of cycles in $\tilde{P}(p)P^{-1}$ for each permutation $P$ and its faulty realization $\tilde{P}(p)$. The next step is to average the formula for the error factor over all realizations of $\tilde{P}$ and finally over all permutations $P$.

Let us start by considering any $k$ element cycle $C$ (constructed by $k-1$ swaps) from permutation $P$, name its unperfect realization $\tilde{P}(p)$ and the number of omitted swaps as $l(p)$. We also want to emphasize that in order to maintain the connection between the number of cycles in qubit permutation $\tilde{P}$ and its trace, we treat each node unmoved by $\tilde{P}$ as a 1-cycle. According to the optimal implementation of the cycle presented in Fig. 9 each node in the cycle $C$ except two is connected with two "neighbouring" nodes via swaps. Thus, one can enumerate nodes in cycle $C$, starting from 0, starting from the node which is moved only by the second layer. Then tag the node connected to it by second-layer swap as 1, next tag the node connected to node 1 by first-layer swap as 2, next tag the node connected to node 2 in the second layer as 3 and so on, see fig 9.

Notice that in the composition $C^{-1}$ with its faulty realization $D$ the errors in the first layer result in corresponding swaps, each connecting nodes with numbers $2m+1$ and $2m+2$ for some integer $m$, "sandwiched" by the second layer. Moreover, the errors in the second layer can be rewritten as additional swaps, connecting nodes with numbers $2m+2$ and $2m+3$ for some

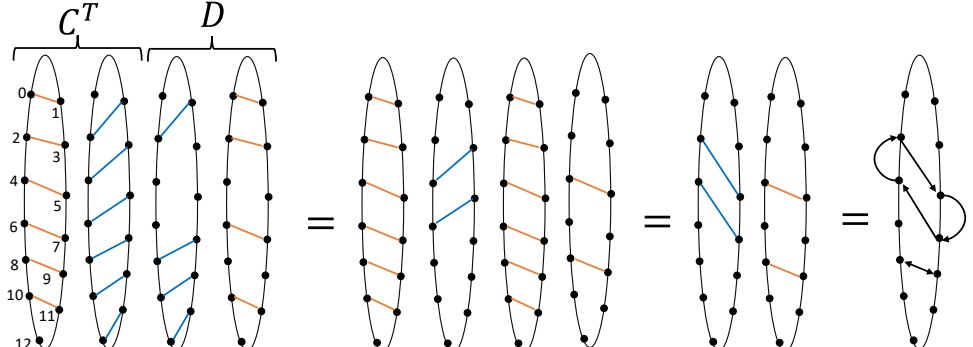

Figure 9: Graphic representation of decomposed 13-cycle $C$ combined with its decomposed faulty realization $D$. Each line corresponds to a swap and each arrow to the shift in the final product. In the first step, the errors from the first (blue) layer are combined to the extra swaps and the errors in the second (red) layer in $D$ are decomposed by additional outer swaps. Next, the "perfect outer layer" moves the inner swaps, as described in the text and the outer errors complete them to cycles.

integer $m$, canceling appropriate swaps in the perfect second layer (see fig 9 first equality). In the next step, we combine two perfect second layers with the errors from the first layers sandwiched between them. This results in the shift of each extra first-layer swap from the pair nodes $2m+1$, $2m+2$ into the pair of nodes $2m$, $2m+3$ (see fig 9 second equality, blue lines). Finally, we multiply those transformed extra first-layer swaps by the swaps corresponding to the second-layer errors. Notice that each of the modified first-layer errors lowers the number of cycles in the product $C^\top D$ by one since each of them combines two 1-cycles into one 2-cycle. Moreover, each second-layer error connects two cycles into one larger cycle, because, by the previous discussion, it cannot cancel the first-layer swap.

Thus, we established a linear dependence between the number of omitted swaps $l(p)$ and the number of cycles in $D(p)$: $m(C,p)$.

$$m(C,p) = k - l(p) = k - \sum_l \binom{k-1}{l} p^l (1-p)^l = k - (k-1)p = k(1-p) + p\,.$$

To consider the entire permutation $P$ one must add the $m(C,p)$ for all cycles $C$ in $P$.

$$m(P,p) = \sum_{cycles} k(1-p) + p = L(1-p) + cp\,, \tag{D.6}$$

where $c$ is a number of cycles in $P$. Therefore, the average error factorizes in the following way:

$$\delta_{full}(p) = \langle\langle \text{Tr}\big[\Pi^\top \tilde{\Pi}(p)\big]^2 \rangle_p\rangle_P = \langle\langle 4^{m(P,p)} \rangle_p\rangle_P = \langle\langle \prod_{C_i} 4^{m(C_i,p)} \rangle_p\rangle_P\,.$$

Where the inner average is taken over all false realizations of $\tilde{P}(p)$ of $P$ and the outer average is taken over all permutations $P$ on $L$ elements, and in the last step we used the decomposition of permutation $P$ into its cycles $C_i$. Let us start with the calculation of the middle average. Consider one chosen cycle $C$ of $k$ elements, assuming that the positions of omitted swaps are uncorrelated:

$$\langle 4^{m(C,p)} \rangle_p = \sum_{l=0}^{k-1} \binom{k-1}{l} p^l (1-p)^{k-1-l} 4^{k-l} = 4(4-3p)^{k-1}\,. \tag{D.7}$$

Because swaps in different cycles are uncorrelated we may combine the above calculation into an average of a trace of the entire permutation:

$$\delta_{full}(p) = \langle\langle \text{Tr}\big[\Pi^\top \tilde{\Pi}(p)\big]^2\rangle_p\rangle_P = \langle\prod_{C_i} 4(4-3p)^{k_i-1}\rangle_P = \langle 4^c(4-3p)^{L-c}\rangle_P = \dots, \qquad \text{(D.8)}$$

where $c$ is the number of cycles in the permutation. Because the distribution of number of cycles is given by Stirling numbers, we follow the calculations:

$$\delta_{full}(p) = \sum_{c=1}^{L} \frac{1}{L!}\begin{bmatrix}L\\c\end{bmatrix} 4^c(4-3p)^{L-c} = 4^{f(p)L}\sum_{c=1}^{L}\frac{1}{L!}\begin{bmatrix}L\\c\end{bmatrix} 4^{g(p)c},$$

with

$$f(p) = \log_4(4-3p) \approx 1 - \frac{3p}{4\log(4)} - \frac{9p^2}{32\log(4)} + O(p^3),$$

and

$$g(p) = 1 - f(p) = \frac{3p}{4\log(4)} + \frac{9p^2}{32\log(4)} + O(p^3).$$

So, one clearly sees that for small $p$ both $f(p)$ and $g(p)$ can be assumed to be linear with very small corrections for reasonably small $p$. For example, for the extremal value of $p = 0.5$, the corrections from all nonlinear terms sum to only $0.068\cdots$, whereas the $f(p)$ takes a value $f(0.5) \approx 0.661$.

Since from the study general bounds (D.4), we know that (at least for small $p$) the fidelity should decay exponentially as $\delta_{full}(p) \approx 4^L e^{-\alpha(L)Lp}$, using this assumption we might derive the value of $\alpha(L)$ using Eq. (D.8). Notice that by the above mentioned *anzatz* we can express parameter $\alpha(L)$ as:

$$
\begin{aligned}
\alpha(L) &\approx -\frac{\partial \log(\delta(p))}{\partial p}\Big|_{p=0}\frac{1}{L} = -\frac{\frac{\partial \delta(p)}{\partial p}\big|_{p=0}}{\delta(0)}\frac{1}{L}\\
&= -\frac{1}{4^L L}\left(4^{f(0)L}\log(4)Lf'(p)|_{p=0}\sum_{c=1}^{L}\frac{1}{L!}\begin{bmatrix}L\\c\end{bmatrix} 4^{g(0)c} + 4^{f(0)}\sum_{c=1}^{L}\frac{1}{L!}\begin{bmatrix}L\\c\end{bmatrix} 4^{g(0)c}\log(4)g'(p)|_{p=0}c\right)\\
&= -\frac{\log(4)}{4^L L}\left(4^L Lf'(0)\sum_{c=1}^{L}\frac{1}{L!}\begin{bmatrix}L\\c\end{bmatrix} + 4^L\sum_{c=1}^{n}\frac{1}{L!}\begin{bmatrix}L\\c\end{bmatrix} g'(0)c\right)\\
&= -\log(4)\left(f'(0) + g'(0)\frac{1}{n}\sum_{c=1}^{L}\frac{1}{L!}\begin{bmatrix}L\\c\end{bmatrix} c\right) = -\log(4)\left(f'(0) + g'(0)\frac{H_L}{L}\right)\\
&\approx -\log(4)\left(f'(0) + g'(0)\frac{\log L + \gamma}{L}\right) = \frac{3}{4}\left(1 - \frac{\log L + \gamma}{L}\right).
\end{aligned}
$$

Thus for large $n$ (and reasonably small p), up to the corrections of order $O(\log(L)/L)$, the decay constant is equal $\alpha = 3/4$, so the average error factor $\delta(p)$ behaves as

$$\delta_{full}(p) \approx 4^L e^{-\frac{3}{4}pL\left(1-\frac{\log L+\gamma}{L}\right)} \approx 4^L e^{-\frac{3}{4}pL}. \qquad \text{(D.9)}$$

The quadratic and higher corrections in $p$ for this formula originate from the higher orders of $f(p)$ and $g(p)$ expansion and a mixed term which decays as $O(\log(L)/L)$, thus as we have shown can be neglected to a good approximation.

# E   Other architectures

From now on we focus our attention on simple models of physical architectures: 1D architecture of all qubits connected in line, 2D architecture with qubits placed in a square lattice and 3D and higher dimensional architectures with qubits placed in a cubic lattice.

For each of those scenarios, we discuss the optimal, or almost optimal way to decompose permutations given the connectivity. Next given the error model discussed above, we calculate the formula for error factor $\delta$ from the formula of fidelity (B.61), which for small errors is close to exact and for larger errors gives a lower bound.

In the quantum volume test, one has the freedom to modify and optimize the circuit as long as its overall action on quantum states is the same. Thus at the end of each subsection, we also present an estimated bound for error factor $\delta(p)$ given that the permutations are not explicitly implemented but each of them is separately "optimized" during implementation.

## E.1   Linear architecture

In this section, we discuss linear architecture with only nearest-neighbour interactions. Thus, the permutation must be implemented as a composition of swap operators $S_{i,i+1}$ between $i$ and $i+1$ qubits. One way of decomposition of a given permutation into such swaps is the brick sort algorithm (also known as left-right sort or parallel bubble sort) [41]. This approach guarantees the minimal number of swaps involved and gives an upper bound for a number of layers (equal to a number of qubits). Hence, if necessary, we will focus on this decomposition and apply it to the generic permutation.

The distribution of the number of swaps $w_s$ necessary to implement a typical permutation $\sigma$ of $n$ elements is known as Mahonian distribution [42], and very quickly approximates the Gaussian distribution, with average $\overline{w}_n$ and variance $\text{Var}(w_s)$ equal:

$$\overline{w}_L = \frac{L(L-1)}{4}, \qquad \text{Var}(w_L) = \frac{2L^3 + 3L^2 - 5L}{72}, \tag{E.1}$$

see [42]. The number of layers in each permutation could be bounded by the longest path in this permutation. This can be further lower bounded if we consider only right-moving paths. Then number of permutations of $n$ elements, with such path of length $k$ number is given by $T(n,k) = \max_i(\sigma_i - i) = k!((k+1)^{n-k} - k^{n-k})$. The average length of the longest right-moving path, using Ramanujan $P$-function, is given by [57]

$$P(L) = \sum_{k=0}^{L-1} \frac{kT(L,k)}{k!} \approx (L-1) - \sqrt{\frac{(L-1)\pi}{2}} + O(1). \tag{E.2}$$

Thus the number of layers one needs to use for the decomposition of standard permutation tends to the maximal number of layers - $L$.

From a computational point of view, the simplest efficient way to obtain a decomposition of a given permutation $\sigma$ using a minimal number of swaps and a small number of layers is to utilize a brick sort algorithm also known as odd-even sort or parallel bubble sort [41]. One just needs to sort the permutation save the sequence of swaps and then apply it layer by layer in the inverse order on qudits. Since a brick sort may not be optimal, it gives an upper bound on a number. Moreover, since it is a parallelization of bubble sort, it can also be upper bounded by $n$ layers.

Later we will use one more distribution $T(L,k)$ [58] of the number of signed paths of length $k$ in all permutations of $n$ elements:

$$T(L,k) = (L-|k|)(L-1)! \,.$$

By convention, we will describe right-moving paths with positive $k$ and left-moving paths with negative $k$. Using this distribution, one can, for example, calculate the average (unsigned) length of the path averaged over all permutations of $L$ elements

$$\overline{n}_L = \frac{1}{L!L}\sum_k |k|T(L,k) = \frac{L^2-1}{3L}. \tag{E.3}$$

For general architecture, the direct connection between the number of omitted swaps in $\tilde{P}(p)$ and the cycles of $\tilde{P}(p)P^{-1}$ is unfortunately no longer valid. This is so because, contrary to the implementation of fully connected architecture, the omitted swaps can be stacked on top of each other, so that new errors do not create new cycles in $\tilde{P}(p)P^{-1}$ but modify the existing one.

Nevertheless, below we try to calculate the average Fidelity decay in scenarios where the errors are so sparse, later called *sparse error regime*, that the abovementioned phenomena occur with negligible frequency. One sees, that when there is on average more than one error in each layer: $p \geq \overline{l}/\overline{w} \approx 1/L$, the discussion below is almost certainly not valid. However, when there are on average one or fewer errors in the implementation of standard permutation: $p \leq 1/\overline{w} \approx 1/L^2$, there is a large chance that we did not exploit our assumptions too drastically.

We also emphasize that since further from sparse error regimes new omitted swaps have a smaller chance of reducing the error factor $\delta(p)$, thus the approximation we are deriving is in fact a lower bound for the error factor $\delta(p)$.

Thus, for sparse error regime (suitable small $p$), we assume that number of cycles in $\tilde{P}$, $m(P,p)$, is given by

$$m(P,p) \approx L - l(p),$$

where $L$ is the number of qubits, and $l(p)$ the number of omitted swaps. Hence, one clearly sees that for $l \approx L$, so average one error in a layer, $p \approx 1/L$, this approximation cannot be true. The average overall realizations for one permutation gives

$$\langle 4^{m(P,p)} \rangle_p \approx \sum_{l=0}^{w} \binom{w}{l} p^l (1-p)^{w-l} 4^{L-l} = 4^L \left(1 - \frac{3p}{4}\right)^w = 4^L 4^{f(p)w}. \tag{E.4}$$

Where

$$f(p) = \log_4(1-3p/4) \approx -\frac{3p}{4\log(4)} - \frac{9p^2}{32\log(4)} + O(p^3),$$

so exactly as before taking our assumptions of small $p$ into account we may safely assume that $f(p)$ is linear.

Since from the study of general bounds (D.4), we know that (at least for small $p$) the error factor should behave as $\delta(p)_{1D} \approx 4^L e^{-\alpha(L)L^2 p}$. using this assumption we might derive the value of $\alpha(L)$ from the Eq. (E.4). This results in

$$
\begin{aligned}
\alpha(L) &\approx -\frac{\partial \log(\delta(p))}{\partial p}\bigg|_{p=0}\frac{1}{L^2} = -\frac{\frac{\partial \delta(p)}{\partial p}\big|_{p=0}}{\delta(0)}\frac{1}{L^2}\\
&= -\frac{1}{4^L L^2}\left(4^L \sum_{w=0}^{L(L-1)/2} 4^{f(0)w}\rho(w)\log(4)f'(p)|_{p=0}w\right)\\
&= -\frac{\log(4)f'(0)}{L^2}\left(\sum_{w=0}^{L(L-1)/2}\rho(w)w\right) = -\frac{\log(4)f'(0)}{L^2}\frac{L(L-1)}{4} = \frac{3}{16}\left(1-\frac{1}{L}\right).
\end{aligned}
$$

Where $\rho(w)$ are weights coming from the Mahonian distribution, and so is the expectation value of $\overline{w}$. Thus, for large $L$ and appropriately small $p \ll \frac{1}{L}$, up to the corrections of order

$O(1/L)$, the decay constant is equal $\alpha = 3/16$, so the average fidelity behaves as

$$\delta(p)_{1D} \gtrsim 4^L e^{-\frac{3}{16}pL^2\left(1-\frac{1}{L}\right)} \approx 4^L e^{-\frac{3}{16}pL^2}. \tag{E.5}$$

The above derivation can be generalized in a straight-forward way into

$$\delta(p)_{1D} \gtrsim 4^L e^{-\frac{3}{4}p\overline{w}_L}, \tag{E.6}$$

where $\overline{w}_L$ is the average number of swaps in permutations of $L$ elements for a given architecture.

### E.1.1 Upper bound of error factor $\delta$ for optimized permutation in 1D

As mentioned at the beginning of the section, in real-life quantum volume tests the permutations do not need to be exactly implemented. Only the performance of the entire circuit needs to agree. This does not mean, however, that we cannot leverage the information about the average behavior of permutations and about 1D architecture.

In particular, if permutation brings together a pair of distant qubits on which a 2-qubit gate is applied, then no matter how good one's transpiler is, those qubits need to be moved to each other. Therefore to calculate *general minimal* number of swaps to implement "optimized" permutation we first need to calculate the average distance $\overline{dist}$ between two qubits that are moved to each other.

Consider a permutation $P$ represented by a permutation matrix $\Pi(P)$. Then a pair of elements that are moved to each other corresponds to a pair of rows in $\Pi(P)$ and the distance between those elements is the distance between columns in which there are 1 in those two rows. Because we consider the average over all permutations, without loss of generality we can consider first two rows. Moreover, for each distance - each occupied pair of columns - the number of permutations is exactly the same. Hence the formula for the distance is given by:

$$\overline{dist} = \frac{\sum_{i\neq j=1}^{L}|i-j|}{\sum_{i\neq j=1}^{L}1} = \frac{\sum_{ij}^{L}|i-j|}{L(L-1)} = \frac{\frac{1}{3}L\left(L^2-1\right)}{L(L-1)} = \frac{1}{3}(L+1).$$

Finally, for each pair of cubits, the sum of distances travelled by two of those must be at least the original distance between them. Thus we can give a strict and always true bound for the average number of swaps for 1D architecture

$$\overline{w}_L \geq \lfloor L/2\rfloor \left(\overline{dist}-1\right) \approx \frac{L(L-2)}{6}, \tag{E.7}$$

which, using (E.5) gives the following upper bound for error factor for small $p$:

$$\delta(p) < 4^L e^{-\frac{1}{8}pL(L-2)} \approx 4^L e^{-\frac{1}{8}pL^2}. \tag{E.8}$$

### E.2 Square cube and hypercube architectures

The most natural generalization of linear architecture is square architecture, where the $L$ qubits are arranged in a square of size $\sqrt{L} \times \sqrt{L}$, and the swaps are allowed between nearest neighbours in both axes. In this subsection, we will discuss such a case and along the way, we generalize presented results for higher dimensional cubes.

For such an arrangement, one can easily derive a lower bound of an average number of necessary swaps $\overline{w}_L$ and layers $\overline{t}_L$ to implement permutations of $L$ elements on a square.

Firstly let's consider the average number of layers $\overline{t}_L$. The average length of the longest right-moving path in one dimension was (asymptotically) given by

$(L-1)+\sqrt{(L-1)\pi/2}+O(1)$, and in two dimensions elements can also move in "top-bottom" direction, effectively skipping $\sqrt{L}$ elements at once. Thus the average longest right-bottom-path, and therefore the average number of layers to implement a permutation, is bounded from below by

$$\overline{t}_L \geq \frac{L-1}{\sqrt{L}} + \sqrt{\frac{(L-1)\pi}{2L}} + O(L^{-1/2}) = O(\sqrt{L}). \tag{E.9}$$

In the case of higher-dimensional cubes of dimension $d$ the above expression generalizes to

$$\overline{t}_L \geq \frac{L-1}{L^{1-\frac{1}{d}}} + \frac{\sqrt{(L-1)\pi}}{\sqrt{2}L^{1-\frac{1}{d}}} + O(L^{-\frac{d-1}{d}}) = O(L^{\frac{1}{d}}), \tag{E.10}$$

by the analogous arguments.

To study a lower bound on the average number of swaps $\overline{w}_L$ necessary to implement permutations of $L$ elements let us notice that each swap can decrease the sum length of all paths in a permutation ($\sum_{i=1}^{L} n_{i,P}^{2D}$) by at most 2. Thus the number of swaps in permutation $\pi$: $w_\pi$ cannot be smaller than half of the sum of the length of all paths. Averaging this relation over all permutations of $n$ elements we obtain:

$$\overline{w}_L \geq \frac{1}{L!} \sum_{P \in \Sigma_L} \frac{1}{2} \sum_{i=1}^{L} n_{i,P}^{2D}. \tag{E.11}$$

In the next step, we once again bound the length of the path in $2D$ architecture, by its length on a line which, using (E.3), gives us:

$$\overline{w}_L \geq \frac{1}{L!} \sum_{P \in \Sigma_L} \frac{1}{2} \sum_{i=1}^{L} \frac{1}{\sqrt{L}} n_{i,P} = \frac{1}{2}\sqrt{L}\,\overline{n}_L = \frac{L^2-1}{6\sqrt{L}} = O(L^{\frac{3}{2}}). \tag{E.12}$$

The generalization into higher-dimensional cubes of dimension $d$ gives us

$$\overline{w}_L \geq \frac{1}{2} L^{\frac{d-1}{d}}\,\overline{n}_L = \frac{L^2-1}{6L^{1-\frac{1}{d}}} = O(L^{1+\frac{1}{d}}). \tag{E.13}$$

### E.2.1 Hypercube sorting

Below we describe an algorithm which gives an efficient upper bound for the average necessary number of layers and swaps simultaneously to implement a permutation using hypercube architectures. The implementation of the algorithm in the Python language is provided in the github repository. The main idea of this algorithm is aligned with [40] theorem 4.3, but our derivation is self-sustained and fully structural thanks to the properties of discussed architectures. Moreover, due to explicit construction, we can argue simultaneously about both the necessary number of swaps and layers to implement a permutation.

As we already argued the problem of implementing a permutation is equivalent to the problem of sorting the inverse of that permutation, thus we focus on the second one for convenience.

Let us start with a square. If all elements from each column were in the correct columns, one would just perform brick sort in each column, see 10, thus the maximal number of layers would be equal $\sqrt{L}$, the maximal number of swaps $\sqrt{L} \times \frac{\sqrt{L}(\sqrt{L}-1)}{2} \approx L^{3/2}/2$. If some elements are in the wrong column, but in each row, there are elements from all columns, one must first sort the rows, again by brick sort. Thus placing all elements in the correct columns, and simplifying the problem to the previous one, see 10b.

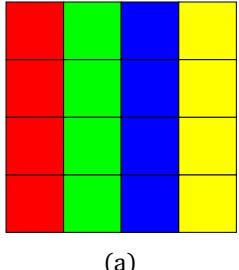
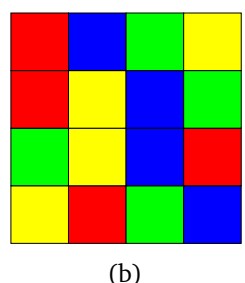
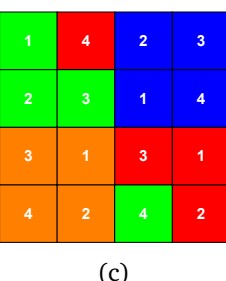

(a)                              (b)                              (c)

Figure 10: In the picture (a), elements on a square lattice with correct columns, the original column for each element is denoted by its colour. In the middle picture (b), columns are incorrect, however, there is only one colour in each row, so one could sort it by brick sort. In the picture (c), a most complex case with appropriate marking is presented.

In general, however, elements which should be placed in one column are randomly scattered through the entire square, see 10c. We claim that the general case can be reduced to the one described in the above paragraph. One may mark each element in a square by natural numbers from 1 to $\sqrt{L}$ in such a way, that in each column there are all marks (without repetitions) and elements which should be in one column are marked with all marks (without repetitions). The proof that such enumeration is always possible, and an explicit algorithm for such enumeration, is placed at the end of the section for clarity. Then after sorting by marks in each column, using brick sort, the elements with marker $i$ end up in row number $i$, so by the properties of enumeration we reduced the problem to the above-described.

Overall to sort a 2D square of $L$ elements we thus did 3 times parallel brick sort - one in columns one in rows and again one in columns - giving maximally $3\sqrt{L}$ layers and no more than $3L^{3/2}/2$ swaps.

Now we can iteratively generalize this method to hypercubes of $L$ elements. Similarly as above we mark the elements in $d$ dimensional hypercube by natural numbers form 1 to $L^{\frac{1}{d}}$, such that in each 1 dimensional column there are all markers and the elements which should be in one column are marked by all the marks. Then we perform sorting over marks in each 1 dimensional column. After this sorting, by the properties of marking, in each $L^{\frac{1}{d}}$ of $d-1$ dimensional stratums there are elements with the same marks, thus in each stratum there are elements which should be placed in all columns. The next step is to perform recursive sorting in those $d-1$ dimensional stratums, after which all elements are in the correct columns, so we finish sorting by applying brick sort in each column separately.

It is a simple proof by induction that such a way of sorting gives the following bound on the maximal, hence also the average number of layers and swaps:

$$\bar{t}_L \le L^{\frac{1}{d}}(2d-1), \quad \text{and} \quad \overline{w}_L < L^{\frac{d+1}{d}}\left(d - \frac{1}{2}\right). \tag{E.14}$$

The only missing part in the above-described algorithm is the proof that appropriate enumerations can always be done. For the general case of hypercubes, we prove the following statement

**Theorem 1** *Let $p$ be a permutation of $m \times j$ elements organized in the rectangle of size $m \times j$. Then there always exists a way to mark all elements of $p$ by the numbers from 1 to $j$ such that in each column are all markers from 1 to $j$ and the set of elements originated from each column has all the markers form 1 to $j$.*

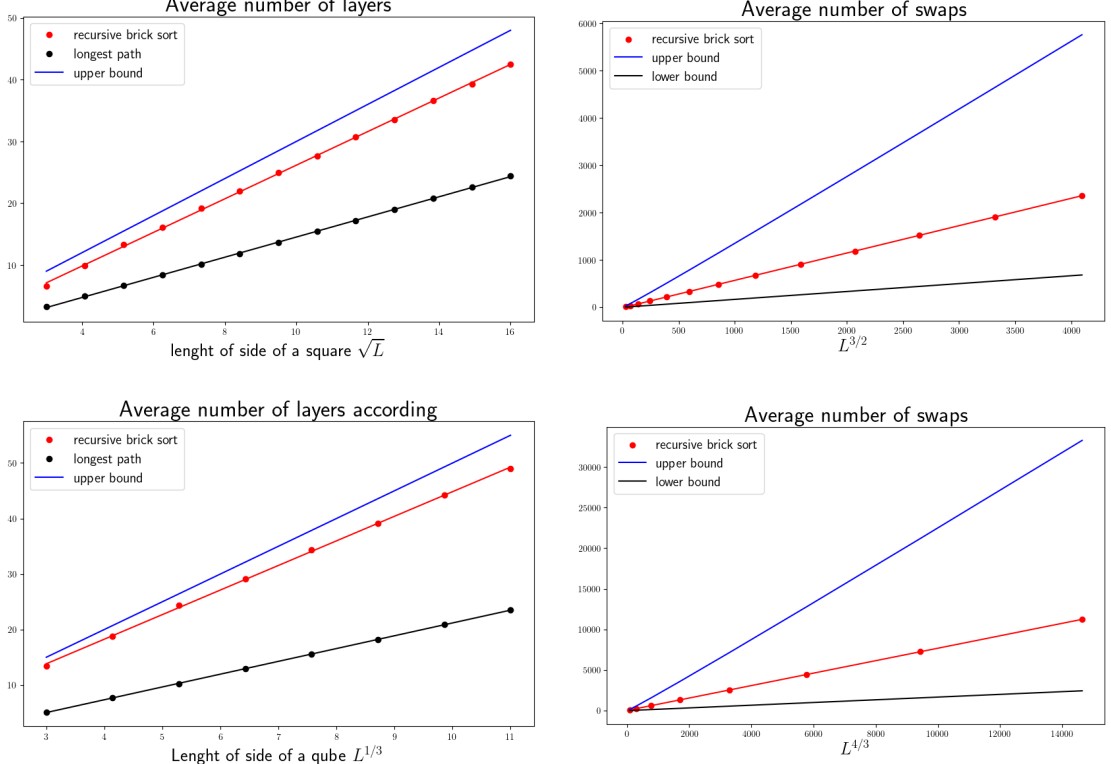

Figure 11: Plot of the average number of layers (left) and swaps (right) to implement a permutation by the recursive version of the brick sort discussed above for the square of $n$ elements and cube of $n$ elements. The upper bound follows directly from the discussion within the algorithm (E.14) and the lower bound from the discussion of the average length of paths (E.10)(E.13). Each point was obtained as an average over 1000 trials.

**Proof:**   If $m = 1$ all marking numbers are 1 so the theorem is trivially satisfied, so in the following, we assume $m > 1$. Each permutation $p$ can be decomposed into a finite number of transpositions, thus we prove the theorem by induction over consecutive transpositions.

As a first inductive step let us notice that if $p$ is an identity, there is a straightforward way of marking: the marker of each element is just its position in the column. Next, we assume that there was some correct marking for the permutation $p$ and that the permutation $p'$ differs from $p$ by one extra transposition of elements.

If those elements had the same marks in $p$, or they originated from the same column, marking for $p$ is also a valid marking for $p'$. Moreover, if those elements belong to the same column, but originated from different rows, the valid marking for $p'$ differs from the marking for $p$ by the same transposition.

Now, we come to the last, most complicated, case, where the transposition changing $p$ into $p'$ mixes the elements which originated from different columns, are in different columns, and have different markings.

To construct the new marking for $p'$ we first copy all the marks, except those with values the same as for swapped elements. Then we start to rewrite the marks from one of two columns with exchanged elements. First, we exchange those marks which are the same as those of swapped elements in one of the columns. This move resolved the conflict with marks for the first swapped element without affecting the property that in each column there are all marks. However, this exchange of marks created a new error for elements which originated from the

same column as the one with just exchanged marks.

So next we look for the element with the same mark from this set, identify its position in this column and swap the two marks of interest. This swap resolved the above-mentioned error but potentially created a new one, thus we further proceeded in the discussed manner. Because the number of columns is limited, this procedure finishes under a finite number of steps with all the conflicts resolved. If some marks for $p$ weren't transcribed into marks for $p'$ in this process, the final step is to copy them without any changes. Therefore we constructed the proper marking for the permutation $p'$. □

The above theorem guarantees the existence of correct marking not only for permutations on a square architecture (for $m = j = \sqrt{L}$) but also on a hypercube ($m = L^{\frac{1}{d}}$, $j = L^{\frac{1-d}{d}}$) since all dimensions except the first (column) one can be flattened without affecting the properties of marking.

Hence, according to to (E.6) we obtain the following bounds for error rate $\delta(p)$:

$$\delta(p)_{2D} \approx 4^L e^{-\frac{9}{8}pL^{\frac{3}{2}}}, \quad \text{and} \quad \delta(p)_{dD} \approx 4^L e^{-\frac{3}{4}pL^{\frac{d+1}{d}}\left(d-\frac{1}{2}\right)}. \tag{E.15}$$

### E.2.2 Upper bound of error factor $\delta$ for optimized permutation in 2D and higher dimensions

For square architecture, similarly, as for 1D architecture, we may lower bound the minimal necessary number of swaps by calculating the distance between a pair of qubits that are brought together. All the arguments regarding the average over permutations from 1D case hold still, but now we consider displacement in two dimensions so the average distance between the pair of qubits is given by:

$$\overline{dist}_{2D} = \frac{\sum_{(i_1,i_2)\neq(j_1,j_2)=1}^{\sqrt{L}} |i_1 - j_1| + |i_2 - j_2|}{\sum_{(i_1,i_2)\neq(j_1,j_2)=1}^{\sqrt{L}}} = \frac{\frac{2}{3}L^{3/2}(L-1)}{L(L-1)} = \frac{2}{3}L^{1/2}. \tag{E.16}$$

This calculation can be easily generalized to $d$ dimensional case:

$$\overline{dist}_{dD} = \frac{\frac{d}{3}L^{2-\frac{1}{d}}\left(L^{\frac{2}{d}}-1\right)}{L(L-1)} \approx \frac{d}{3}L^{1/d}. \tag{E.17}$$

Hence same as (E.7) we can derive a lower bound on the necessary number of swaps to implement an "optimized" permutation. Which gives the following upper bounds for the error factor:

$$\delta(p)_{2D} \lesssim 4^L e^{-\frac{1}{4}pL^{\frac{3}{2}}}, \quad \text{and} \quad \delta(p)_{dD} \lesssim 4^L e^{-\frac{d}{8}pL^{\frac{d+1}{d}}}. \tag{E.18}$$

## F  Connection between the heavy-output frequency and the fidelity

In this section, we first present the definition and design of a heavy-output frequency test. Then we provide additional evidence for the connection between heavy output frequency and fidelity [44].

For a given random quantum circuit $U$ and an input state $|\psi_0\rangle$, the output state is denoted as $|\Psi\rangle = U|\psi_0\rangle$. The basis states measured with a probability greater than the median $p_{\text{med}}$ of all probabilities are named *heavy outputs*, constituting the *heavy output subspace* represented by $H_U$.

$$H_U = \{|m\rangle \text{ s.t. } p_m > p_{\text{med}}\}, \tag{F.1}$$

where $p_{\boldsymbol{m}} = |\langle \Psi | \boldsymbol{m} \rangle|^2$ denotes the probability of measuring a basis state $|\boldsymbol{m}\rangle$. The heavy-output probability $h_U$ is defined as

$$h_U = \sum_{|\boldsymbol{m}\rangle \in H_U} p_{\boldsymbol{m}}. \tag{F.2}$$

This concept is useful for benchmarking quantum computers. Under specific assumptions, it has been demonstrated that no classical algorithm can identify heavy outputs with a probability greater than 2/3 [14]. Consequently, a quantum device's ability to exceed this probability threshold of 2/3 may signify a quantum advantage in sampling, making it a passing criterion in the Quantum Volume test [23]. In the QV test, the heavy output subspace is identified by the classical simulation of the quantum circuit $U$. A real quantum device executes a corresponding faulty circuit $\tilde{U}$ that generates an outcome state $|\tilde{\Psi}\rangle = \tilde{U} |\psi_0\rangle$. The probabilities of the basis states of the heavy output subspace, determined by classical simulation of a quantum circuit, are measured leading to the faulty heavy output frequency $h_{\tilde{U}}$, which reads

$$h_{\tilde{U}} = \sum_{|\boldsymbol{m}\rangle \in H_U} \tilde{p}_{\boldsymbol{m}}. \tag{F.3}$$

For an ideal, error-free circuit, the asymptotic average heavy output frequency approaches $h_U \rightarrow (1 + \log(2))/2 \approx 0.85$, compared to 0.5 for a completely depolarized device [14].

It turns out that for the discussed types of errors and architectures of the quantum volume circuit, the connection between the fidelity and heavy output frequency can also be stated as a simple function, which is later discussed numerically. The relation is given by the linear rescaling such that the lower and upper bounds for both quantities coincide:

$$\mathcal{F} = 1 - \frac{2^L - 1}{2^L} \frac{h_U - h_{\tilde{U}}}{h_U - \frac{1}{2}}, \tag{F.4}$$

where $h_U$ is the average value of heavy output frequency obtained for the ideal scenario with no errors. The numerical evidence supporting this claim is presented in Figure 12, where we studied the influence of the parameters and architecture.

We note that the same relation was obtained in the case of *global* depolarizing channel [44], which suggests that this simple behaviour is general at least for isotropic noise. The fact that standard approximations of heavy output frequency, which stream from the behaviour of fidelity, repeatedly provided the expected results [23] [44] additionally supports this claim.

Despite the observed linear relation, deriving an analytical expression for the average heavy output frequency in the presence of noise remains challenging. The expression for the average heavy output frequency can be obtained in the spirit of Eq. (A.3)

$$\overline{h_{\tilde{U}}} = \sum_{\boldsymbol{m}} \langle \boldsymbol{m}\boldsymbol{m} \| \overline{\mathcal{P}_H(U_T \cdots U_1) \otimes \mathcal{P}_H(U_T \cdots U_1) \overleftarrow{\prod_{\tau}} \left[ \tilde{U}_\tau \otimes \tilde{U}_\tau^* \right]} \| \psi_0 \psi_0 \rangle, \tag{F.5}$$

where $|\psi_0\rangle$ is the chosen input state, the $\mathcal{P}_H$ denotes the projection on the appropriate heavy output subspace and in general is dependent both on the gates inside the circuit and the chosen input state. Due to this interrelation, the averages cannot be factorized even for the uncorrelated errors. Moreover, the unitarity in the circuit is lost as well making the analytical calculations unattainable in general scenarios.

The numerical simulations were made using Julia as the programming language. Libraries such as *LinearAlgebra*, *Random*, *Statistics* and *Permutations* were used. All calculations were made with a precision of a 64-bit floating point. An example of the code used to compute both average fidelity and heavy output frequency in the 1D architecture can be found in the github repository. The data points were obtained for several layers considering $12 \leq T \leq 20$. Each

point was obtained through an average over some number of iterations and for each architecture we considered several values of $\alpha$ and $p$, to spread the points in the curve. The values of all these parameters are shown in Tables 2 and 3. The uncertainty associated with each point (and both quantities $\overline{\mathcal{F}}$ and $\overline{h_{\tilde{U}}}$) is the standard error, calculated by dividing the standard deviation by the square root of the number of iterations. The maximum error obtained was of the order $\sim 10^{-2}$, so we chose to omit the error bars in Figure 12.

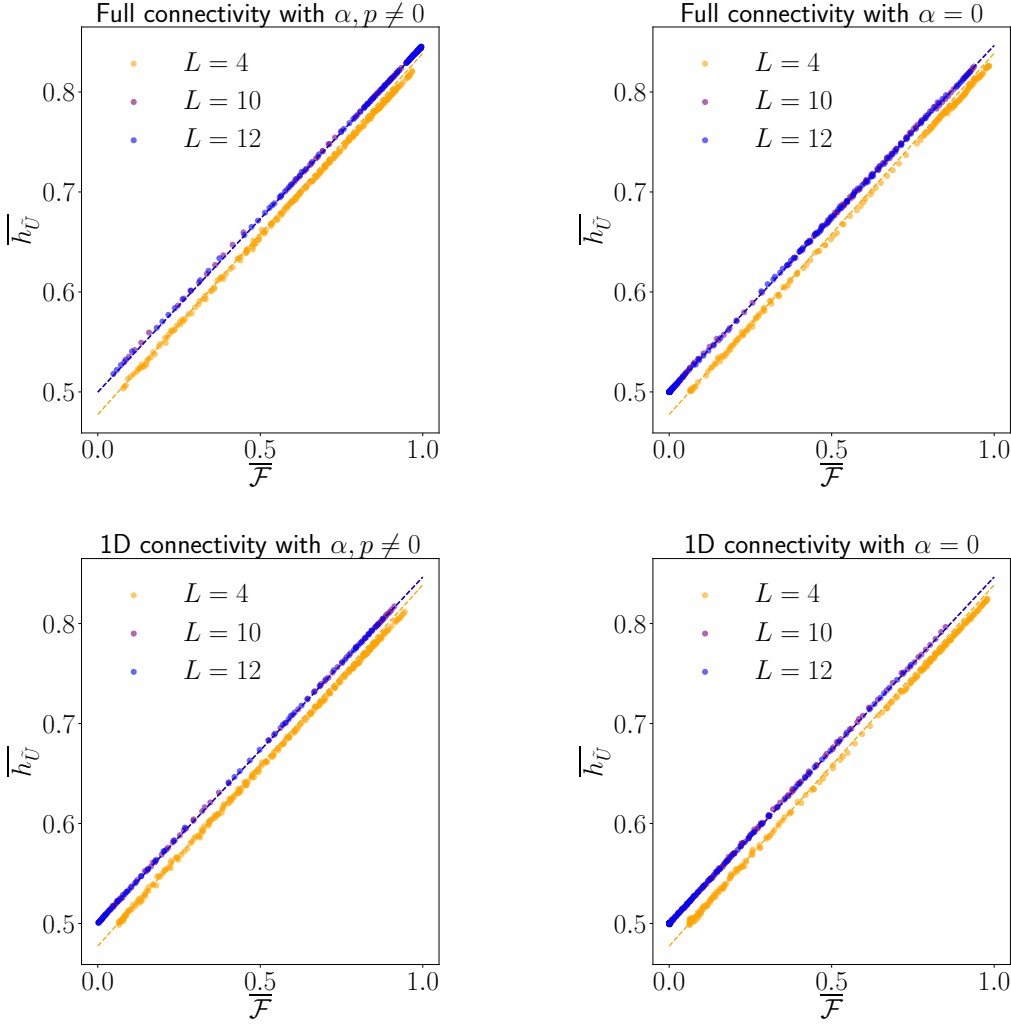

Figure 12: Plot of the average faulty heavy output frequency as a function of the average fidelity for the two architectures considered. We observe a linear relation between fidelity and heavy output.

Table 2: Parameters used in the numerical simulations for both architectures, in the case of $\alpha = 0$. A total number of twenty values of $p$ were used.

| Architecture | $L$ | N° of iterations | $\alpha$ | $p$ |
|---|---|---|---|---|
| Full connectivity | 4 | 10000 | | |
| | 10 | 20000 | | |
| | 12 | 20000 | 0 | $n \times 10^{-j}$ with $n \in \{1, 2, \ldots, 9\}$, $j \in \{1, 2, 3\}$ and $p \leq 0.2$ |
| 1D connectivity | 4 | 5000 | | |
| | 10 | 2000 | | |
| | 12 | 1000 | | |

Table 3:  Parameters used in the numerical simulations for both architectures, in the case of $\alpha, p \neq 0$. For all systems sizes and full connectivity, and $L = 4$ and 1D connectivity, a total of twenty-seven pairs of $(\alpha, p)$ were used. For the other two cases, only eight pairs were considered.

| Architecture | $L$ | Nº of iterations | $(\alpha, p)$ |
|---|---|---|---|
| Full connectivity | 4 | 5000 | $(0.008, 0.0048), (0.04, 0.008),$ $(0.08, 0.008)$ and $(\alpha^*, c\alpha^*),$ with $\alpha^* \in \{0.001, 0.002, 0.003,$ $0.0045\}$ and $c \in \{2, 5, 7, 10, 20, 30\}$ |
| | 10 | 2000 | $(0.008, 0.0048), (0.04, 0.008),$ $(0.08, 0.008)$ and $(\alpha^*, c\alpha^*),$ with $\alpha^* \in \{0.001, 0.002, 0.003,$ $0.0045\}$ and $c \in \{2, 5, 7, 10, 20, 30\}$ |
| | 12 | 1000 | $(0.008, 0.0048), (0.04, 0.008),$ $(0.08, 0.008)$ and $(\alpha^*, c\alpha^*),$ with $\alpha^* \in \{0.001, 0.002, 0.003,$ $0.0045\}$ and $c \in \{2, 5, 7, 10, 20, 30\}$ |
| 1D connectivity | 4 | 5000 | $(0.008, 0.0048), (0.04, 0.008),$ $(0.08, 0.008)$ and $(\alpha^*, c\alpha^*),$ with $\alpha^* \in \{0.001, 0.002, 0.003,$ $0.0045\}$ and $c \in \{2, 5, 7, 10, 20, 30\}$ |
| | 10 | 2000 | $(0.001, 0.002), (0.002, 0.0012),$ $(0.003, 0.0006), (0.003, 0.006),$ $(0.0045, 0.00045), (0.008, 0.0048),$ $(0.04, 0.008), (0.08, 0.008)$ |
| | 12 | 1000 | $(0.001, 0.002), (0.002, 0.0012),$ $(0.003, 0.0006), (0.003, 0.006),$ $(0.0045, 0.00045), (0.008, 0.0048),$ $(0.04, 0.008), (0.08, 0.008)$ |

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
