# Peer review of "Fidelity decay and error accumulation in random quantum circuits"

_SciPost Physics, doi:SciPost Phys. 19, 013 (2025)_

## Round 3 · Referee Report · Anonymous (Referee 1) · 2025-2-10

Strengths

1- Strong mathematical description and proof 2- The intuition to create a solvable model that captures the relevant features of the actual quantum computation. 3- Well written.

Weaknesses

1- I miss clear motivation and prospects (personal opinion) 2- The paper should be more clear on the impact of this research, in particular in the context of existing literature.

Report

See report

Attachment

Recommendation

Ask for minor revision

  • validity: good
  • significance: good
  • originality: good
  • clarity: high
  • formatting: good
  • grammar: excellent

Author:  Nadir Samos  on 2025-06-02  [id 5533]

(in reply to Report 1 on 2025-02-10)

Dear Referee,
please see document attached.
With our best wishes
The authors

Attachment:

Reply_Referee1.pdf

---

## Round 3 · Referee Report · Anonymous (Referee 2) · 2025-4-3

Strengths

1.- The writing is clear, with well-structured sections, and the calculations are complete without missing steps. I enjoyed reading the paper.
2.- The authors perform detailed, non-trivial analytical calculations for the fidelity of the solvable model and provide examples for different device architectures. In this sense, the study is very complete.
3.- The techniques and calculations used may be of independent interest.

Weaknesses

1.- The relevance and possible applications of this work are not entirely clear to me. The exponential fidelity decay is expected, and the chosen noise model is just one among many possibilities. Moreover, introducing global Haar random unitaries in the solvable model makes it more arbitrary. There may exist other, possibly simpler, random circuits whose predictions numerically match those of the original circuit.

Report

This paper investigates how noise affects random quantum circuits in quantum computing. The authors focus on two types of random circuits:

1.- The original circuit, composed of layers of random two-qubit gates followed by a random permutation of qubits. These circuits are used in quantum volume benchmarking.

2.- The solvable circuit, composed of layers of random two-qubit gates, followed by a global Haar unitary, a random permutation, and another global Haar unitary.

They study the impact of two main sources of noise on these circuits: faulty two-qubit gates and faulty permutations. To quantify the effects of these errors, the paper analyzes the entanglement fidelity between the target and noisy circuits, averaged over all random instances of these circuits. The authors derive an analytical expression for the fidelity decay in the solvable model, showing its dependence on the number of qubits, the number of circuit layers, and the noise parameters. The results show that even for small systems, the target and noisy circuits can be quite different. Numerical simulations indicate that the analytical predictions for the solvable model closely match the numerical results for the better-motivated original model, suggesting that both circuits exhibit similar fidelity decay trends under noise.

Overall, I believe this study was well-conducted and represents a solid piece of research that deserves publication in SciPost Physics, primarily due to its technical contributions.

Requested changes

1.- The motivation and relevance of this work were not entirely clear to me. While I understand that these circuits are related to quantum volume, this is just one possible noise model for quantum volume circuits, and quantum volume itself is only one of several benchmarking methods for quantum computers. Are there any other possible applications?
2.- How does this work relate to previous research? Have similar studies been conducted before?
3.- Why did you choose this error model? Why did you choose this particular form for the solvable model? The global Haar random unitaries seem to be included purely for convenience. Could a simpler circuit yield the same predictions?
4.- I find the text around equation (B13) unclear.

Recommendation

Ask for minor revision

  • validity: good
  • significance: good
  • originality: good
  • clarity: high
  • formatting: good
  • grammar: excellent

Author:  Nadir Samos  on 2025-06-02  [id 5534]

(in reply to Report 2 on 2025-04-03)

Dear Referee,
please see document attached.
With our best wishes
The authors

Attachment:

Reply_Referee2v2.pdf

Author:  Nadir Samos  on 2025-06-02  [id 5532]

(in reply to Report 2 on 2025-04-03)

Dear Referee,
please see document attached.
With our best wishes
The authors

Attachment:

Reply_Referee2.pdf

---

## Round 4 · Author Response

We are very grateful for your consideration.
We sincerely appreciate the time and effort both Referees have dedicated to reviewing our work, in particular for the level of commitment required for thoughtful analysis of the technical results. Their detailed comments and critical assessments have been invaluable in refining our manuscript. We firmly believe that we adequately addressed their feedback. Primarily, we tried to clarify the argumentation for the solvable model used for analytical calculations, and extended the motivation and background for our work, the problems raised by both referees.
With these revisions, we believe that the new version of our paper now meets the standards for publication in SciPost.
With our best wishes,
The Authors

---

## Round 4 · List of Changes

1.- Added the following paragraph in the introduction that emphasizes the motivation and usefulness of our work.
``In the context of quantum chaos and dynamic phase transitions, quantum fidelity is termed Loschmidt echo[25,31], measuring the extent to which a complex system is recovered after applying an imperfect (perturbed) time-reversal. In the framework of time-independent Hamiltonians, the behaviour of the Loschmidt echo is well understood for single-particle quantum systems whose dynamics are fully chaotic in the classical limit: it typically exhibits an initial parabolic decay, followed by an exponential one, and eventually saturates [31]. This pattern has also been observed in many-body systems [32], and similar behaviour is expected in systems governed by time-dependent Hamiltonians, such as quantum circuits [33]. While a quantitative understanding is valuable in its own right, it becomes particularly pertinent in light of the technological relevance of quantum circuits.’'
2- Expanded the paragraph in the introduction, where we discuss the circuit with random permutations emphasizing the versatility of RQCs:
``For example, the brick-wall circuit—consisting of a sequential alternation of leftward and rightward single-qubit shifts—is a paradigmatic model of local random quantum circuits (RQCs)[4]. It has been studied extensively in the context of information spreading [5], thermalization[6], and measurement-induced phase transitions [1]. In contrast, random permutations have been used to model black hole dynamics with non-local interactions[7,8], to establish bounds on entanglement generation[9,10], to study pseudo-randomness and unitary $k$-designs—ensembles that reproduce Haar-random statistics up to the $k$-th moment [11,12]—and to investigate quantum complexity [13], among other applications.''
3- To better motivate the explicit noise model chosen for our calculations, we extended the last paragraph in Section 2.1:
``Since the two-qubit gates are already random, noise must be modelled as a random deviation from the uniform sampling defined by the Haar measure. To this end, to preserve the symmetry with which the gates were sampled we consider that each random unitary $u_{r,r'}$ is independently perturbed by unstructured noise: $\tilde{u}{r,r'} = e^{i\alpha h$}} u_{r,r', where $h_{r,r'}$ is drawn from the Gaussian Unitary Ensemble (GUE), and $\alpha \geq 0$ controls the noise strength. Notably, the ability to model noise independently in both the permutations and the two-qubit gates provides significant flexibility and control in our framework.’'
4- Added a sentence to the final paragraph of the conclusions highlighting a possible research direction beyond the quantum-computing-inspired topics already discussed.
``In addition, it could be interesting to explore the connection between average fidelity in the generic models considered and out-of-time order correlators (OTOCs), inspired by the known relation between the Loschmidt echo and OTOCs in systems governed by time-independent Hamiltonians [48].’’
5- Modified discussion around Eq, (B13):
``It is convenient to exploit the fact that the GUE measure is invariant under unitary transformations. In particular, this implies that the eigenvectors of $H$ do not favour any specific direction in Hilbert space. Therefore, the unitary matrix $U$ that diagonalizes $e^{i\alpha H}$, namely
6.- Corrected typos throughout the text

---

## Editorial Decision

published